# Genetic analysis of the Arabidopsis TIR1/AFB auxin receptors reveals both overlapping and specialized functions

Michael J Prigge[1], Matthieu Platre[2], Nikita Kadakia[1†], Yi Zhang[1‡], Kathleen Greenham[1§], Whitnie Szutu[1], Bipin Kumar Pandey[3], Rahul Arvind Bhosale[3], Malcolm J Bennett[3], Wolfgang Busch[2], Mark Estelle[1*]

[1]Section of Cell and Developmental Biology, University of California San Diego, La Jolla, United States; [2]Plant Molecular and Cellular Biology Laboratory and Integrative Biology Laboratory, Salk Institute for Biological Studies, La Jolla, United States; [3]Plant and Crop Sciences, School of Biosciences, University of Nottingham, Nottingham, United Kingdom

**Abstract** The TIR1/AFB auxin co-receptors mediate diverse responses to the plant hormone auxin. The Arabidopsis genome encodes six TIR1/AFB proteins representing three of the four clades that were established prior to angiosperm radiation. To determine the role of these proteins in plant development we performed an extensive genetic analysis involving the generation and characterization of all possible multiply-mutant lines. We find that loss of all six TIR1/AFB proteins results in early embryo defects and eventually seed abortion, and yet a single wild-type allele of *TIR1* or *AFB2* is sufficient to support growth throughout development. Our analysis reveals extensive functional overlap between even the most distantly related *TIR1/AFB* genes except for *AFB1*. Surprisingly, *AFB1* has a specialized function in rapid auxin-dependent inhibition of root growth and early phase of root gravitropism. This activity may be related to a difference in subcellular localization compared to the other members of the family.

**\*For correspondence:**
mestelle@ucsd.edu

**Present address:** [†]University of California, Riverside School of Medicine, Riverside, United States; [‡]Department of Human Genetics, University of California, Los Angeles, Los Angeles, United States; [§]Department of Plant and Microbial Biology, University of Minnesota, Saint Paul, United States

**Competing interests:** The authors declare that no competing interests exist.

## Introduction

The phytohormone auxin regulates diverse processes throughout the entire plant life cycle. Auxin acts as a signal to promote cell differentiation during morphogenetic events such as embryogenesis, root development, and shoot organ formation. Auxin also mediates responses to environmental cues such as light, gravity, water availability, and pathogens. Auxin regulation of transcription involves three families of proteins; AUXIN RESPONSE FACTOR (ARF) transcription factors, Aux/IAA transcriptional repressors, and TRANSPORT INHIBITOR RESPONSE1 (TIR1)/AUXIN-SIGNALING F-BOX (AFB) proteins. Auxins, of which indole-3-acetic acid (IAA) is the predominant natural form, are perceived by a co-receptor complex consisting of TIR1/AFB and Aux/IAA proteins. Formation of the co-receptor complex leads to degradation of the Aux/IAA protein and activation of ARF-dependent transcription (Reviewed in *Lavy and Estelle, 2016*). In addition to this established pathway, recent studies demonstrate that the TIR1/AFB proteins are required for very rapid auxin responses in the root and in developing root hairs that are independent of transcription (*Dindas et al., 2018*; *Fendrych et al., 2018*). The details of TIR1/AFB function in these rapid responses are currently unknown, but in the root, the response is thought to be important for early events in gravitropism.

Members of the TIR1/AFB protein family are encoded by three pairs of paralogs in the *Arabidopsis thaliana* genome. Each protein contains an amino-terminal F-Box followed by eighteen leucine-rich repeats (LRRs). Only *tir1*, *afb2*, and *afb5* mutants have been identified in forward-genetic screens (*Ruegger et al., 1997*; *Ruegger et al., 1998*; *Alonso et al., 2003*; *Walsh et al., 2006*;

*Parry et al., 2009*), but reverse-genetic analyses revealed functional redundancies between *TIR1*, *AFB2*, and *AFB3* as well as between *AFB4* and *AFB5* (*Dharmasiri et al., 2005*; *Prigge et al., 2016*).

Gene duplication events provide the primary source material for the evolution of biological innovation. In plants, whole genome duplication (WGD) events have been especially important with events preceding the radiation of several key plant lineages including seed plants, flowering plants, and core eudicots (*Jaillon et al., 2007*; *Jiao et al., 2011*; *Clark and Donoghue, 2018*). Following duplication, the paralogs are often redundant, allowing one copy to degenerate into a pseudogene (*Lynch and Conery, 2000*). In Arabidopsis, the average half-life of a duplicate gene has been estimated at 17.3 million years (*Lynch and Conery, 2003*). In many cases, however, both duplicates are retained for one or a combination of reasons (reviewed in *Panchy et al., 2016*). Occasionally, one of the paralogs evolves a novel function (neofunctionalization), but often the two paralogs fulfill different aspects (enzymatically, temporally, or spatially) of the role of the ancestral gene (subfunctionalization). Following subfunctionalization, there may be changes in selective pressure allowing each paralog to evolve specialized functions without affecting functions carried out by the other paralog. This mechanism likely played a prominent role in the evolution of plant gene families and, in turn, in the radiation and diversification of land plants.

The *TIR1/AFB*, *Aux/IAA*, and *ARF* gene families expanded during land plant evolution after the divergence of bryophytes and vascular plants (*Remington et al., 2004*; *Rensing et al., 2008*; *Mutte et al., 2018*). Because auxin has a central role in many important adaptations that occurred during land plant evolution, such as vascular development, lateral root formation, and organ polarity; it seems likely that the acquisition of new roles for auxin was enabled by the duplication and diversification of these three gene families. Here we present the comprehensive genetic analysis of the *TIR1/AFB* gene family of Arabidopsis which revealed extensive functional overlap between even distantly related members as well as an essential role for the TIR1/AFB pathway in early embryos. In contrast the AFB1 protein appears to have adopted a special role in a rapid auxin response in the root.

## Results

### Major lineages of auxin receptors diverged prior to the fern–seed plant split

To better understand the timeframe during which the auxin receptor family diversified, we built upon previous phylogenetic analyses (*Parry et al., 2009*; *Mutte et al., 2018*) with more taxon sampling at key nodes. As shown earlier (*Hori et al., 2014*; *Mutte et al., 2018*), the *TIR1/AFB* genes likely evolved from a gene encoding an F-Box/LRR protein similar to those present in the genomes of extant streptophyte algae. These algal proteins form a sister clade to three distinct land plant F-Box families, the TIR1/AFB auxin receptors, the COI1 jasmonate-Ile (or dinor-OPDA) receptors, and the 'XFB' clade of unknown function conserved in the genomes of mosses and some lycophytes but not in other land plants (*Prigge et al., 2010*; *Bowman et al., 2019*). While the last common ancestors of land plants and of vascular plants had only one *TIR1/AFB* gene, three clades of auxin receptors were established prior to the radiation of euphyllophytes (ferns plus seed plants) over 400 million years ago (*Morris et al., 2018*; *Figure 1—figure supplement 1A*). Another gene duplication event prior to angiosperm radiation split the TIR1/AFB1 clade from the AFB2/AFB3 clade. Receptors from each of the four clades are not retained in the genome of every flowering plant. For example, *AFB6* orthologs are not present in the genomes of core Brassicales species—including Arabidopsis—nor those of Poaceae species including rice and maize. The gene duplication event establishing the distinct TIR1 and AFB1 clades is coincident with the At-β WGD event at the base of Brassicales, while both the AFB2/AFB3 and the AFB4/AFB5 duplication events coincide with the more recent At-α WGD prior to divergence of the Brassicaceae family (*Figure 1—figure supplement 1A*; *Schranz and Mitchell-Olds, 2006*).

One noteworthy aspect of the phylogenetic tree is the pronounced branch-length asymmetry within the TIR1+AFB1 clade (*Figure 1—figure supplement 1*). Since the last common ancestor of *Arabidopsis* (Brassicaceae) and *Tarenaya* (Cleomaceae), the *AFB1* gene has accumulated over three times as many non-synonymous changes as *TIR1* (*Figure 1—figure supplement 1B*) despite being under selection based on the ratio of non-synonymous and synonymous substitutions (*Delker et al.,*

*2010*; *Wright et al., 2017*). AFB1 also differs from the other TIR1/AFBs in that it contains two of three substitutions in the first α-helix of the F-Box that were each previously shown to weaken TIR1's interaction with CUL1 (*Yu et al., 2015*). The substitution with the largest effect, Glu8Lys (equivalent to Glu12Lys in TIR1), appeared shortly after the At-β WGD that produced AFB1, and the Phe14Leu substitution appeared prior to the crown group of the Brassicaceae family (*Figure 1—figure supplement 1C*). Interestingly, *AFB1* orthologs from members of the *Camelina* genus—*C. sativa* (all three homeologs), *C. laxa*, *C. hispida*, and *C. rumelica*—additionally contain the third substitution (*Figure 1—figure supplement 1C*).

## Genetic analysis of the Arabidopsis *TIR1/AFB* gene family revealed extensive functional overlap

Previous studies have assessed the functional overlap between the *TIR1*, *AFB1*, *AFB2* and *AFB3* genes (*Dharmasiri et al., 2005*; *Parry et al., 2009*) and separately between the *AFB4* and *AFB5* genes (*Prigge et al., 2016*). To study the genetic interactions between all members of the family, and to determine the effects of the complete absence of TIR1/AFB-mediated auxin signaling, lines with strong loss-of-function mutations in the six *TIR1/AFB* genes were intercrossed to generate all sixty-three mutant combinations. We used the following alleles *tir1-1, afb1-3, afb2-3, afb3-4, afb4-8*, and *afb5-5* (*Figure 1—figure supplement 2A*; *Ruegger et al., 1998*; *Parry et al., 2009*; *Prigge et al., 2016*). The *tir1-1* allele, which causes an amino acid substitution within the leucine-rich repeat domain of the protein, has been reported to act as a dominant-negative allele (*Dezfulian et al., 2016*; *Wright et al., 2017*). However, we found that the root elongation phenotype of plants heterozygous for the *tir1-1, tir1-10*, and *tir1-9* alleles were not significantly different from each other and each displays a semi-dominant phenotype (*Figure 1—figure supplement 2B*). These results argue against a dominant negative effect for *tir1-1* since neither *tir1-9* or *tir1-10* produce detectable levels of transcript (*Ruegger et al., 1998*; *Parry et al., 2009*). Nevertheless, because it is possible that a dominant-negative effect might be revealed in higher-order mutants and because the *afb2-3* allele may not be a complete null allele (*Parry et al., 2009*), we generated selected mutant combinations also using the *tir1-10* (*Parry et al., 2009*) and the *afb2-1* (*Dharmasiri et al., 2005*) T-DNA insertion alleles. The *afb2-1* allele was introgressed from the Ws-2 background into the Col-0 background through at least eight crosses. For brevity, mutant line names will be simplified such that '*tir1afb25*' corresponds to the *tir1-1 afb2-3 afb5-5* triple mutant line, for example, unless other allele numbers are specified.

The sixty-three mutant combinations displayed a wide range of phenotypes from indistinguishable from wild type to early-embryo lethality (*Figure 1—figure supplement 3*; *Supplementary file 1*). The non-lethal higher-order mutant combinations displayed a cohort of phenotypes associated with mutants defective in auxin signaling: smaller rosettes, reduced inflorescence height, reduced apical dominance, fewer lateral roots, and partially or wholly valveless gynoecia (*Figure 1*; *Figure 1—figure supplement 3*; *Supplementary file 1*). The three viable quintuple mutants—*tir1afb1245, tir1afb1345*, and *afb12345*—had rosettes approximately half the diameter and inflorescences less than half the height of WT Col-0. Despite being smaller, these lines produced approximately twice as many branches as WT (*Figure 1A*; *Figure 1—figure supplement 3*; *Supplementary file 1*). Remarkably, lines retaining only one copy of *TIR1* (*tir1/+ afb12345*) or one copy of *AFB2* (*afb2/+ tir1afb1345*) were viable. The rosettes of these two lines were much smaller than those of WT plants with the *tir1/+ afb12345*'s rosette phenotype being slightly more severe (*Figure 1B*). In contrast, *afb2/+ tir1afb1345* plants developed shorter primary inflorescences and appeared to completely lack apical dominance as all axillary meristems became active upon flowering. The *afb2/+ tir1afb1345* and *tir1/+ afb12345* plants rarely produced seeds. Lines containing the alternate alleles—*afb2-1/+ tir1-10 afb1345* and *tir1-10/+ afb2-1 afb1345*—displayed phenotypes indistinguishable from the corresponding lines (*Figure 1B*).

Auxin plays an important role in many aspects of root development. To begin to assess the role of the *TIR1/AFBs* during root growth, we measured the effect of exogenous IAA on primary root growth in the mutant lines. The responses ranged from indistinguishable from WT to nearly insensitive to 0.5 µM IAA (*Figure 1—figure supplement 3E*), where the roots of lines containing the *tir1* and *afb2* mutations displayed strong IAA resistance (*Dharmasiri et al., 2005*; *Parry et al., 2009*). In addition, we found that the *afb3* and *afb5* mutations had substantial effects on auxin response, while the *afb4* mutation had a more modest effect. The mutant lines also responded similarly to

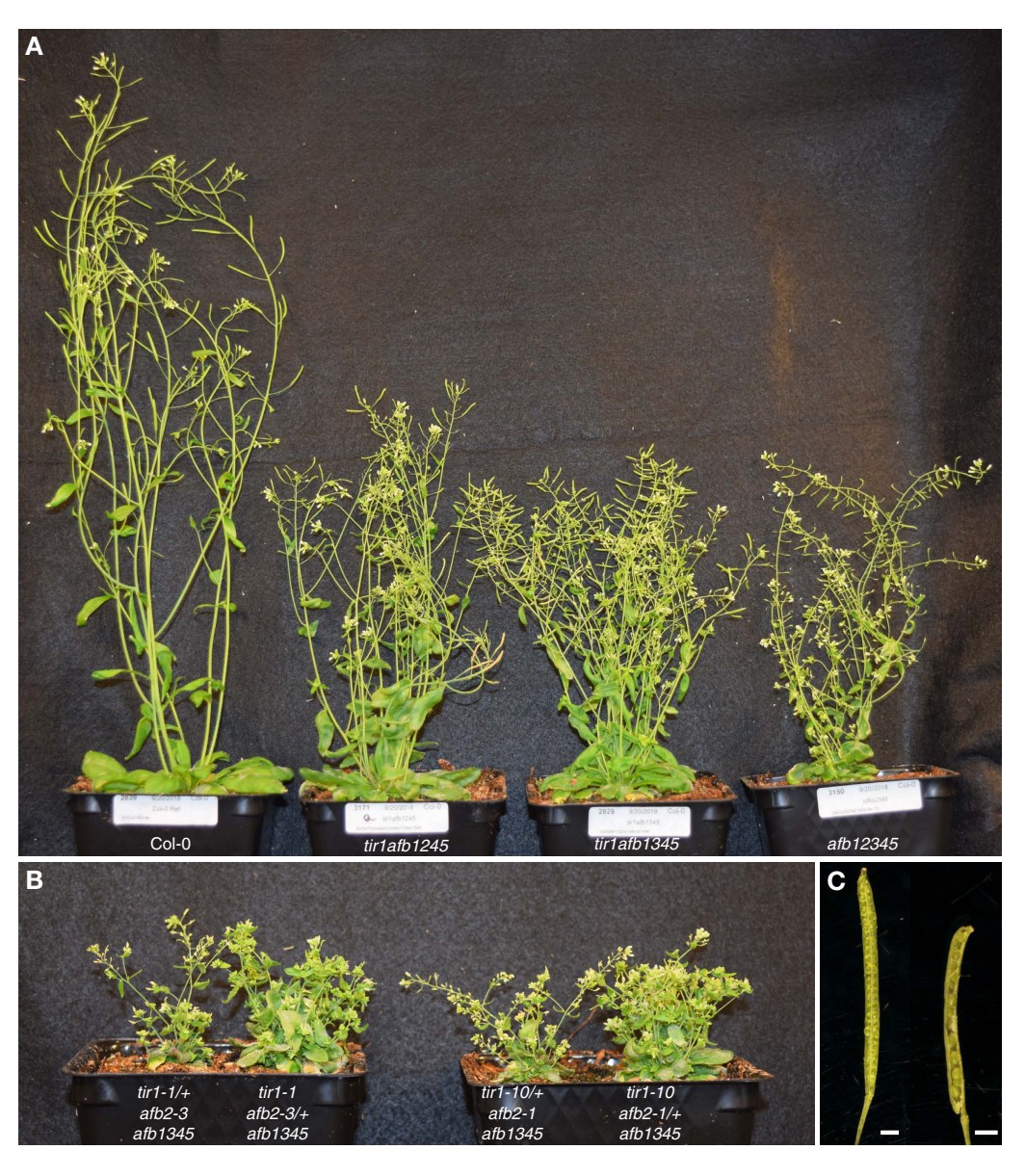

**Figure 1.** *tir1/afb* mutant lines exhibit a range of shoot phenotypes. (**A**) The viable quintuple mutants, *tir1afb1245*, *tir1afb1345*, and *afb12345*, are each approximately half the height of Col-0 WT, but differ in other phenotypes. Note the curved silique tips of the *tir1afb1245* mutant (indicative of gynoecium defects) and the short siliques (due to poor fertility) of the *afb12345* mutant. (**B**) Lines with only one *TIR1+* or one *AFB2+* allele display similar phenotypes regardless of the mutant *tir1* and *afb2* alleles: left to right, *tir1-1/+ afb2-3 afb1345*, *tir1-1 afb2-3/+ afb1345*, *tir1-10/+ afb2-1 afb1345*, and *tir1-10 afb2-1/+ afb1345*. (**C**) Normal siliques (Col-0, left) have two valves containing developing seeds while 32% of *tir1afb1245* siliques have only one. The adaxial half of the valve walls were removed to reveal the developing seeds. Scale bars are 1 mm. Plants were grown for 42 days at 22°C and 16 hr daylength.

The online version of this article includes the following source data and figure supplement(s) for figure 1:

**Figure supplement 1.** TIR1/AFB Phylogeny.
**Figure supplement 2.** Alternate *tir1/afb* alleles.
**Figure supplement 2—source data 1.** Source data for *tir1* dominance test.
**Figure supplement 3.** Shoot and root phenotypes of *tir1/afb* mutants.
**Figure supplement 3—source data 1.** Source data for phenotype measurements.
**Figure supplement 4.** Embryonic root formation in *tir1/afb* mutants.

*Figure 1 continued*

**Figure supplement 4—source data 1.** Source data for seedling phenotypes.

exogenous auxin with respect to lateral root production. The lines more resistant to IAA in the root elongation assay tended to produce fewer lateral roots (*Figure 1—figure supplement 3D and E*).

## Combinatorial mutant analyses revealed roles for *TIR1/AFB* family members except *AFB1*

Each of the *tir1/afb* mutations, except for *afb1*, affected the above-described phenotypes but to varying extents. To appraise the effects of each mutation on several plant phenotypes, we plotted the mean values for each phenotype minus that of the corresponding line without that mutation. Larger effects are indicated by greater deviations from zero. For both the root elongation assay and the induction of lateral root primordia, the *tir1* allele had the largest effect with the *afb2*, *afb5*, *afb3*, and *afb4* mutations having smaller median effects (*Figure 2A and B*). The *afb1* mutation had little or no effect on root elongation but, surprisingly, had an opposite effect on lateral root formation.

Only *tir1* and, to a lesser degree, *afb2* affect rosette diameter in most contexts with the median effects for *afb3*, *afb4*, and *afb5* being very close to zero (*Figure 2C*). However, they have huge

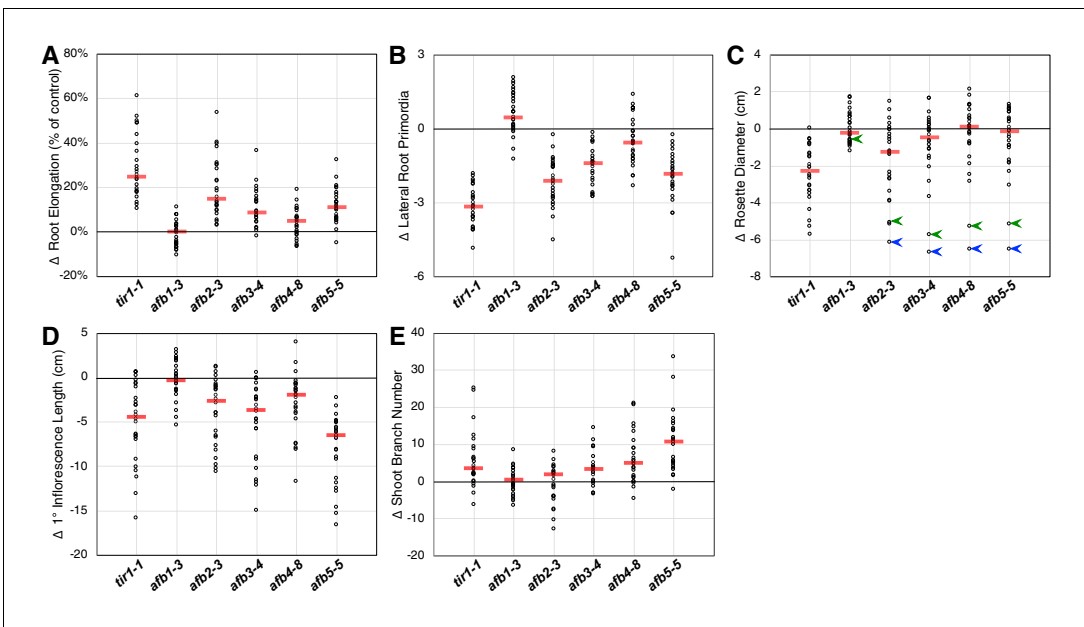

**Figure 2.** Relative *TIR1/AFB* gene effects. For each of the five phenotype measurements (*Figure 1—figure supplement 3*), the normalized mean for each genotype with the given mutation was subtracted from the normalized mean for the corresponding genotype lacking that mutation and plotted (circles). The red bars indicate the median difference attributable to the given mutation. (**A**) Effects of each mutation on IAA-inhibition of root elongation. For each genotype, twelve five-day-old seedlings were transferred and grown for three days on media containing 100 nM IAA, and their average growth was divided by that of twelve seedlings grown on media lacking added auxin. (**B**) Effects of each mutation on auxin-induced lateral root production. Twelve five-day-old seedlings for each genotype were grown for four days on media containing 100 nM IAA and the numbers of emerged lateral roots were counted. (**C**) Effects of each mutation on the average rosette diameters of five 42 day old plants. The blue arrowheads indicate difference in phenotypes between the *afb2345* quadruple mutant and the four triple mutants, and the green arrowheads indicate those for the *afb12345* quintuple mutant and the five quadruple mutants. (**D**) Effects of each mutation on the average height of the primary inflorescences for five 42 day old plants. (**E**) Effects of each mutation on the average number of inflorescence branches (≥1 cm) on five 42 day old plants.

collective effects, where the *afb2345* quadruple mutant is over 6 cm smaller than each of the four triple mutants (blue arrowheads; *Figure 2C*). Consistent with previous reports that *AFB5* plays a key role in regulating inflorescence branching and height (*Prigge et al., 2016*; *Ligerot et al., 2017*), the *afb5* mutation has the largest effect on these phenotypes, although each mutation, except for *afb1*, had some effect (*Figure 2D and E*).

While the *afb1* mutation had minimal effect on most aspects of plant growth, it suppressed the lateral root phenotype of some mutant lines both with and without auxin treatment (*Figure 2B*; below). We found that the *afb1* mutation suppressed the phenotype of both the *afb234* (2.15 ± 0.13 versus 1.75 ± 0.10 lateral roots/cm) and *afb345* triple mutants (3.10 ± 0.13 versus 1.96 ± 0.14 lateral roots/cm) measured after 12 days on media not supplemented with IAA. This behavior was not observed in an otherwise WT background (2.76 ± 0.11 for *afb1* versus 3.23 ± 0.15 lateral roots/cm for Col-0) nor in a *tir1-1* background (1.75 ± 0.08 versus 2.34 ± 0.09 lateral roots/cm for *tir1*). Each of the pairs were significantly different (two-tailed *t*-test, p<0.03).

## Penetrance of *tir1/afb* embryonic root formation defect is temperature sensitive

The *tir1afb23* and *tir1afb123* lines were previously shown to display a variably penetrant phenotype in which seedlings lack roots, lack both roots and hypocotyls, or fail to germinate (*Dharmasiri et al., 2005*; *Parry et al., 2009*). All lines homozygous for both *tir1* and *afb2* plus either *afb3*, *afb4*, or *afb5* display these defects to some degree ranging from 1% in *tir1afb24*% to 99% in *tir1afb1234* (*Supplementary file 1*; *Figure 1—figure supplement 4*).

We had noticed a sizeable difference in the proportion of seedlings lacking roots from different batches of seeds. To test whether the temperature at which the seeds mature affects the penetrance of the rootless seedling phenotype, we grew *tir1afb23*, *tir1afb123*, *tir1afb245* and WT in parallel at 17°C, 20°C, and 23°C and scored the progeny seedling phenotypes (*Figure 1—figure supplement 4*). The penetrance of the phenotype for all three lines was significantly lower at 20°C than at either 17°C or 23°C for all with the exception that the difference with *tir1afb245* at 17°C was not significant using the Fisher's exact test. This suggests that aspects of the auxin regulatory system are sensitive to temperature.

## The *tir1afb12345* mutant line exhibits defects early in embryogenesis

Because *tir1afb235* seedlings were not identified among the progeny of *tir1/+ afb235* or *afb2/+ tir1afb35* plants, we examined developing embryos dissected from the siliques from these lines. Eleven of 41 (27%) and 52 of 213 (24%), respectively, of the embryos from each line lacked cotyledons and had over-proliferated suspensors while the rest had a WT phenotype (*Figure 3A, A'*).

Because all mutant combinations expected to produce one-quarter *tir1afb2345* progeny were either seedling lethal or infertile, we created a transgene that hemizygously complements these phenotype and segregates as a single locus. We assembled the complementing genomic fragments encoding TIR1, AFB2, and AFB5, each carboxy-terminally fused with the coding sequences for different monomeric fluorescent proteins (mOrange2, mCitrine, and mCherry, respectively) concatenated into a single binary plasmid (*Figure 3—figure supplement 1*). This construct was transformed into progeny of *tir1/+ afb5/+ afb1234* plants, backcrossed and selfed to obtain a sextuple mutant background, and two *TIR1/AFB5/AFB2* lines were identified that complemented the sextuple mutant phenotype when hemizygous and segregated as a single locus. Using this approach, one-quarter of the progeny of plants hemizygous for these transgenes display embryo defects, while the complemented siblings are easily identified because they expressed fluorescent TIR1, AFB2, and AFB5 fusion proteins.

The earliest potential difference between sextuple mutants and the complemented siblings is that the initial division of the embryo proper was occasionally displaced from vertical in sextuple 2-cell embryos (3 of 19 were >12° from vertical) compared to complemented sibling embryos (0 of 64), however the average angles from vertical were not significantly different (p=0.32) (*Figure 3B–C* versus 3B'–C', 3N). The third round of divisions in the embryo proper separates the upper and lower tiers with the lower surface of the upper tier typically being slightly convex, and this curvature is significantly more prominent in the sextuple mutants (p=$1.4 \times 10^{-7}$; *Figure 3D* versus 3D', 3O). Later, during the transition from the 8-cell to the 16-cell embryos, nearly all cell divisions in the

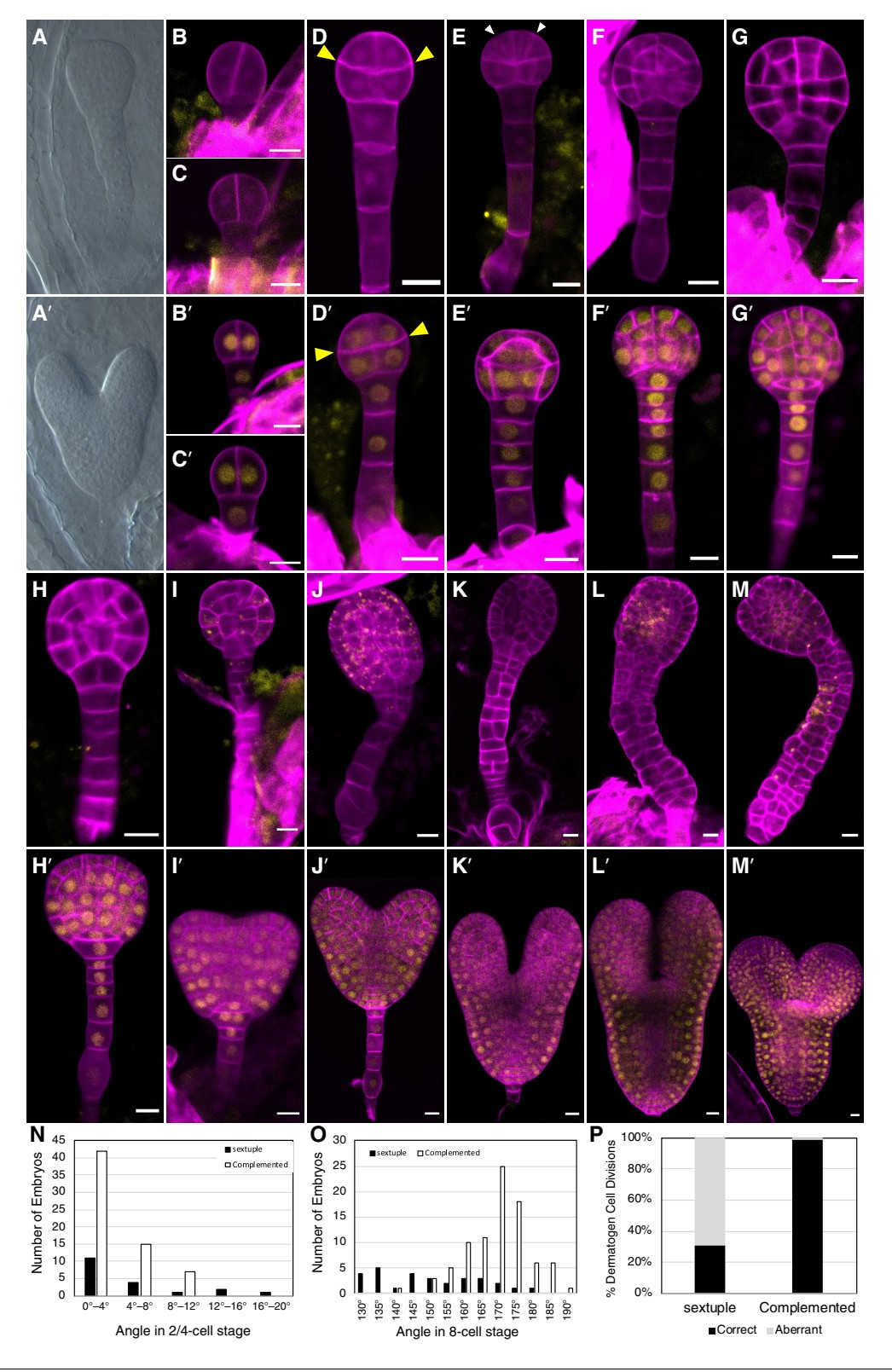

**Figure 3.** Embryo-lethal phenotypes of *tir1/afb* mutant lines. Rows of panels alternate between defective and normal embryos. Approximately one-quarter of the chloral-hydrate-cleared embryos from siliques of *afb2/+ tir1afb35* plants did not produce cotyledon primordia and have over-proliferated suspensors (**A**) while the remaining siblings from the same silique appear normal (**A'**). Embryos from *TIR1/AFB5/AFB2/+ tir1afb12345* plants

*Figure 3 continued on next page*

*Figure 3 continued*

were fixed, stained with SR2200 (cell walls, magenta), and scanned for fluorescence from the AFB2-mCitrine fusion protein (yellow). All were progeny of 'd2' transgenic line and the standard alleles except panels (C and D) contained the *tir1-10* and *afb2-1* alleles and panels H' and I were progeny of plants with the 'd1' transgenic line. The embryos in panels B–M are sextuple mutants lacking mCitrine signal while those in B'–M' are complemented siblings. The embryo stages are 2-cell (B–C, B'–C'), 8-cell (octant; D, D'), 16-cell (dermatogen; E, E'), early globular (F, F'), late globular (G–H, G'–H'), late transition (I, I'), heart (J, J'), torpedo (K–L, K'–L'), and bent cotyledon (M, M'). The yellow cytoplasmic signal in panels (I) through (M) likely represents autofluorescence of senescing cells. (N) Histogram of the angles of the first division plane with 0° defined as perpendicular to a line connecting the upper corners of the hypophysis cell for sextuple (black) and complemented siblings (white). The average difference was not significantly different (p=0.32 from *t*-test, $n = 19$ and 64). (O) Histogram of the angles of lines connecting the upper and lower tiers of octant embryos from side to center to side (indicated by arrowheads in panels D, D'). The means for the sextuple and complemented siblings were 149.1° and 169.1°, respectively, and were significantly different (p=$1.4 \times 10^{-7}$ from *t*-test, $n = 29$ and 84). (P) Bar graph showing the frequencies of normal (periclinal) and aberrant (anticlinal, arrowheads in panel E) divisions in 16-cell embryos. While aberrant divisions were observed in complemented siblings, they were significantly more frequent in sextuple mutants (p=$2.8 \times 10^{-54}$ from Fisher's exact test, $n = 94$ and 437 divisions).

The online version of this article includes the following figure supplement(s) for figure 3:

**Figure supplement 1.** Transgene complementing the *tir1afb12345* sextuple mutant.
**Figure supplement 2.** Appearance of autofluorescence in sextuple mutant embryos.

complemented embryos are oriented periclinally, as in WT embryos. In contrast, 69% of these division are anticlinal in the mutant embryos (*Figure 3E and E'*, 3P). In WT 32-cell stage embryos, the hypophysis cell normally divides asymmetrically to produce the lens-shaped cell which is required for the formation of the embryonic root. This division was delayed in the mutant, and when it occurred, was symmetrical (*Figure 3G–H* and 3G'–H'). Later, the cells of the embryo proper slow or cease dividing and the cells of the suspensor begin to proliferate and invariably produce a radially symmetric terminal phenotype (*Figure 3I–M*, 3I'–M'). Around the stage where complemented siblings are at the bent-cotyledon stage, the cells of the sextuple mutant senesce and seed development is aborted (*Figure 3—figure supplement 2*). Hence, the sextuple mutant reveals the importance of *TIR1/AFB* auxin response machinery from the earliest stages of embryogenesis.

## Expression of embryo-patterning reporters is disrupted in the *tir1afb235* quadruple mutant

To learn more about the early embryo defects, we introgressed marker genes into lines segregating the *tir1afb235* quadruple mutant. This quadruple mutant displays a phenotype indistinguishable from that of the sextuple mutant from at least the dermatogen stage, when quadruple mutants can first be reliably distinguished from non-quadruple mutants (*Figure 4*). Expression of the auxin-responsive marker *DR5rev:3×Venus*-N7 was undetectable in embryos displaying the mutant phenotype (*Figure 4E–G*; compare to *Figure 4A–C*; 0/10 in quadruple mutants and 26/26 in non-mutant siblings) indicating that auxin-regulated transcription through the remaining receptors, AFB4 and AFB1, is minimal at most during embryogenesis. Similarly, the quiescent center marker *WOX5:GFP* was not detected in the mutant embryos (*Figure 4H*), whereas in wild type the reporter is first expressed in the hypophysis prior to its asymmetric division then persists in the quiescent center cells (*Figure 4D*; 0/11 in quadruple mutants and 28/28 in non-mutant siblings).

Auxin efflux reporter *PIN7-Venus* is normally expressed in the suspensor, hypophysis, hypophysis-derived cells, and weakly in protodermal cells of the lower tier (*Figure 4K–L*). In mutant embryos, PIN7-Venus is faintly detectable in these cells in globular-stage embryos. Unexpectedly, the signal is much stronger in protodermal cells of the embryo proper, especially in the lower tier, by the 32-cell stage (*Figure 4O–P*; 7/8 quadruple mutants and 0/14 siblings). The same pattern was observed with the *PIN7-GFP* marker (*Figure 4U*; 8/8 quadruple mutants and 0/30 siblings). The auxin efflux reporter *PIN1-Venus* is initially expressed in a reciprocal pattern to *PIN7-Venus*, in all the cells above the hypophysis except the lower-tier protodermal cells and is later refined to strips from the

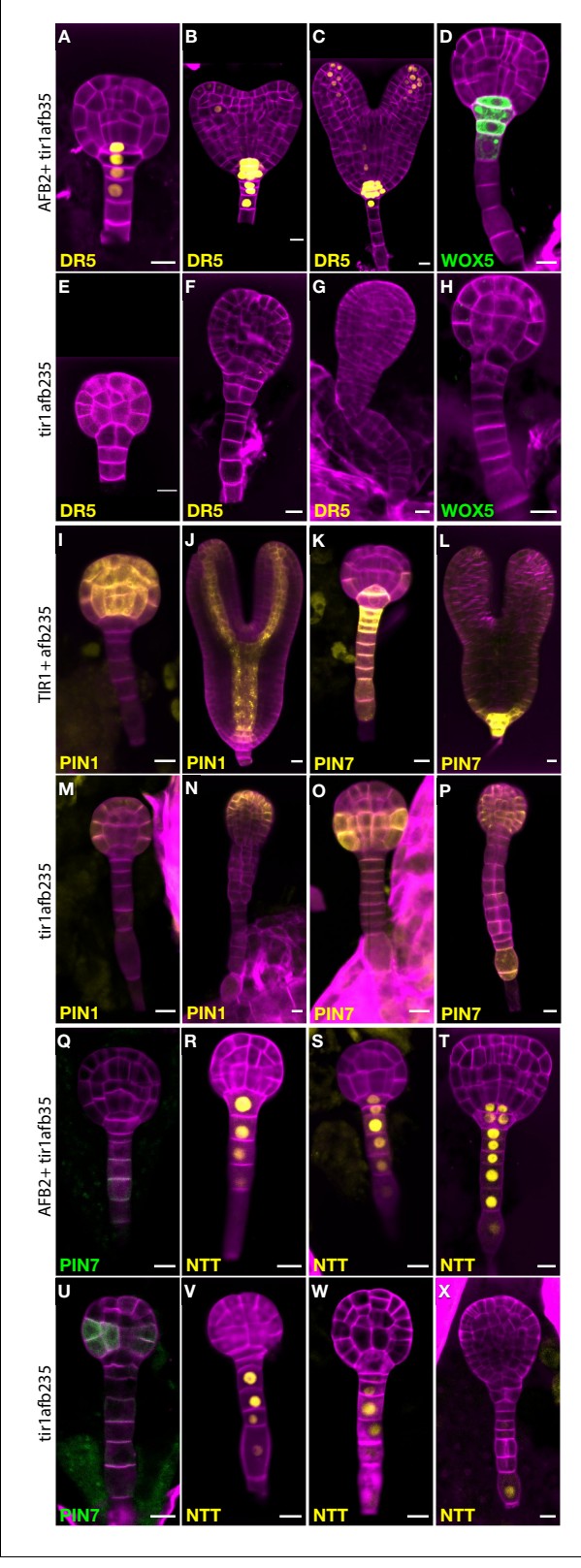

**Figure 4.** Marker gene expression in the *tir1afb235* embryos. Fluorescence in embryos from both *afb2/+ tir1afb35 DR5rev:3×Venus-N7* (A–C, E–G) and *afb2/+ tir1afb35 WOX5:GFP$_{ER}$* (D, H) markers was present in phenotypically normal siblings (A–D) but absent in abnormal (presumed *tir1afb235*) embryos (E–H). Fluorescence in embryos from *tir1/+afb235 PIN1-Venus* plants: normal-phenotype globular embryo (I), normal-phenotype torpedo-stage embryo

*Figure 4 continued on next page*

*Figure 4 continued*

(J), mutant-phenotype globular embryo (M) and later-stage embryo (N). Progeny of *tir1/+ afb235 PIN7-Venus* or *afb2/+ tir1afb35 PIN7-GFP* plants: phenotypically normal globular embryos (K, Q) mutant globular embryos (O, U), and normal (L) and mutant (P) torpedo-stage embryos. Progeny of *afb2/+ tir1afb35 NTT-YPet* plants: normal-phenotype globular- (R–S) and transition- (T) stage embryos, and mutant embryos (V–X). Scale bars: 10 µm.

provascular cells out to the cotyledon tips (*Figure 4I–J*; 12/12 of phenotypically normal embryos). In the mutants, PIN1-Venus signal is reduced and restricted primarily to apical protodermal cells (*Figure 4M–N*; 9/9 quadruple mutants). The *NTT-YPet* marker gene is normally first strongly expressed in 8- to 16-cell embryos in the nuclei of suspensor cells and the hypophysis and persists in the suspensor and the hypophysis-derived cells in later embryo stages (*Figure 4R–T*; 58/59 phenotypically normal embryos) (*Crawford et al., 2015*). In mutants, NTT-YPet appears normally in most suspensor cells, but not always including the hypophysis (2/6 32-cell quadruple mutant embryos had signal above background in the hypophysis), and is progressively lost in the distal suspensor cells before the abnormal lateral cell divisions occur (*Figure 4V–X*; 4/4 late-globular-stage mutants lacked signal in both the hypophysis and the adjacent suspensor cell). This is very similar to NTT-YPet expression in a *monopteros* mutant embryos (*Crawford et al., 2015*).

## Gametophytically expressed TIR1/AFBs do not contribute to gametophytic viability

Because the maternal supply of auxin and the endosperm both play important roles in embryo development, it is possible that female gametophytes lacking auxin receptors would not be viable. Although the incidence of sextuple mutant embryos shows that gametophytically-expressed auxin receptors are not required for viability, it is possible that they contribute to robust transmission. To test the transmission through sextuple mutant megagametophytes and pollen, we carried out reciprocal crosses between wild type (Col-0) and the hemizygously *TIR1/AFB5/AFB2*-complemented sextuple mutant. If the sextuple mutant gametophyte survives and is fertilized, the progeny's embryo lethality would be rescued by wild-type copies of each receptor provided by the Col-0 parent. The $F_1$ progeny were scored for the presence of the transgene to infer the sextuple's transmission rates through both gametophytes (*Supplementary file 2*). The sextuple mutant was transmitted nearly as well as without the complementing transgene as with it through both the pollen (49.4%) and the female gametophyte (47.9%) ($\chi^2$ test p=0.81 and 0.43, respectively). This indicates that gametophytically expressed TIR1/AFBs do not contribute to gametophytic viability.

## Functional TIR1/AFB-mCitrine reporters reveal contrasting patterns of spatio-temporal expression and sub-cellular localizations

To reveal whether differences in expression pattern can account for the relative importance of the TIR1/AFBs in different aspects of growth and development, C-terminal fusions with the bright, relatively fast-maturing, monomeric fluorescent protein mCitrine were produced for each TIR1/AFB protein in the corresponding single mutant background. Each transgene complemented the mutant phenotypes (*Figure 5—figure supplement 1*). The fluorescent signal in the *AFB5-mCitrine* lines was fairly uniformly distributed in shoot apices (*Figure 5F*), while in the *AFB3-mCitrine*, *AFB2-mCitrine*, and *TIR1-mCitrine* lines, fluorescence was more restricted to young primordia and meristem peripheral zones (*Figure 5A and C–D*). Within organ primordia, *TIR1-mCitrine* appears to be strongest in the adaxial domains of the youngest primordia. Signal for the *AFB4-mCitrine* line was barely detectable (*Figure 5E*), while that of *AFB1-mCitrine* was very strong and largely complementary to *TIR1-mCitrine* in that the strongest signal was in abaxial domains and in the stem (*Figure 5B*). The expression patterns in either primary or lateral roots for each *TIR1/AFB* gene except *AFB4* translationally fused to a YFP have been reported previously (*Prigge et al., 2016*; *Wang et al., 2016*; *Rast-Somssich et al., 2017*; *Roychoudhry et al., 2017*). TIR1-, AFB2-, AFB3-, and AFB5-mCitrine signal was uniformly detected throughout the root meristematic region and fainter signal detected in the root cap cells (*Figure 5G, I–J and L*; *Figure 5—figure supplement 2*) as shown previously. AFB1-mCitrine is very highly expressed throughout the root except for the columella, cortex, endodermis, and pericycle of the meristematic region (*Figure 5H*). The expression pattern of an AFB4-mCitrine line (line #3), hypersensitive to the synthetic auxin picloram, was comparable to that of AFB5-

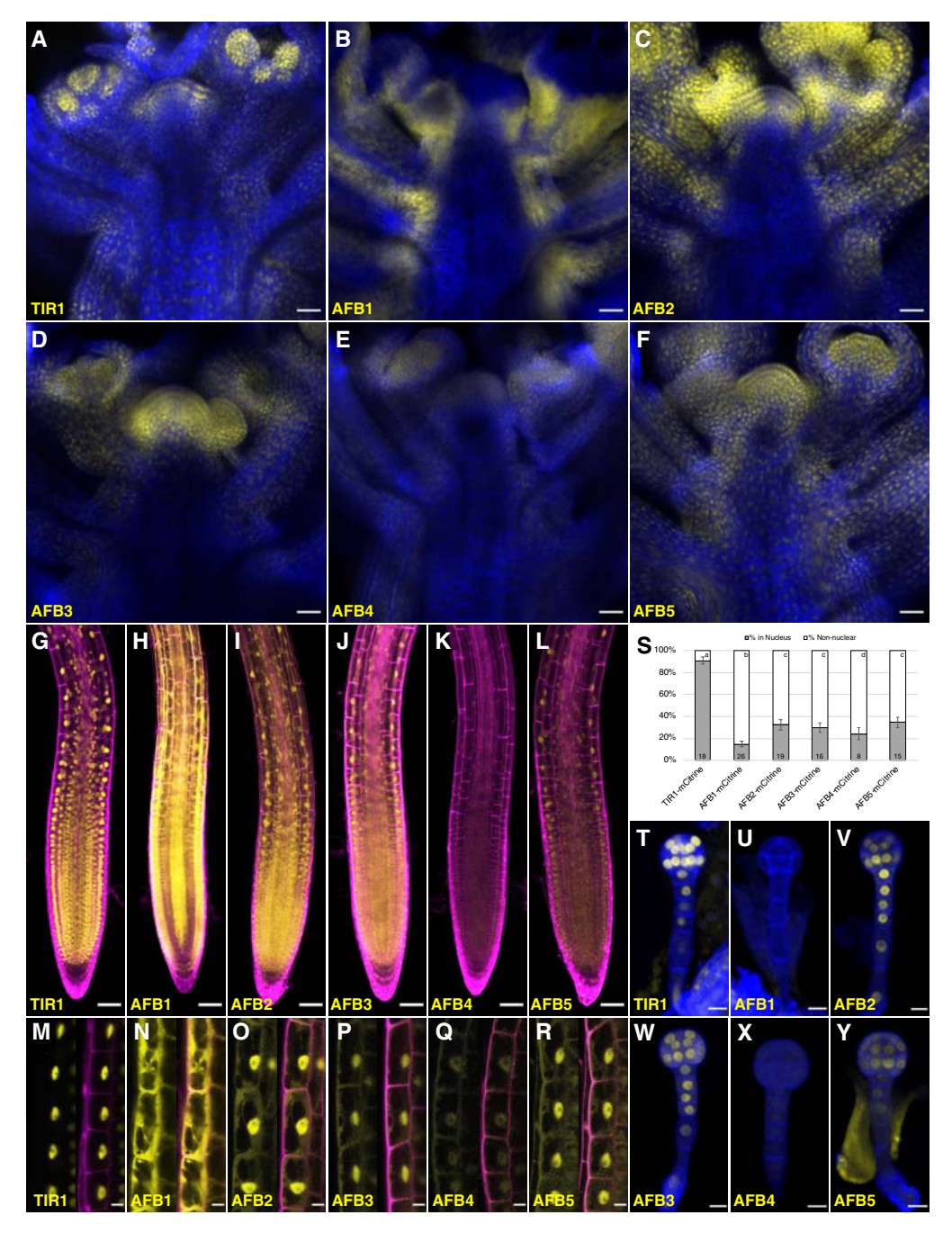

**Figure 5.** Expression of *TIR1/AFB-mCitrine* translational fusions. (A–F) Confocal images of inflorescence apices from 4-week-old plants containing the specified *TIR1/AFB-mCitrine* transgenes. (G–R) Confocal images of roots of 5-day-old seedlings under lower magnification (G–L) or 7-day-old seedlings under higher magnification (M–R). Images in panels (G) and I–L used similar microscope settings while those in panel (H) used less sensitive settings. (S) Plot comparing the relative proportions of mCitrine signal inside the nucleus (gray) and outside the nucleus (white). Cells were imaged and measured for each TIR1/AFB-mCitrine line, and the averages ± standard deviations are shown. For AFB1-mCitrine, F1 hybrids with the *UBQ10:H2B-mTurquoise2* nuclear marker were used so that the nuclei could be delineated (*Figure 5—figure supplement 3*). The numbers in the bars indicate the number of cells measured and the letters distinguish significantly different averages (two-tailed *t*-test p<0.05). (T–Y) Confocal images of dermatogen or early globular embryos. mCitrine signal is shown as yellow in all panels, and cell walls were stained with Calcofluor White M2R (blue; A–F), propidium iodide (magenta; G–R), and SCRI Renaissance

*Figure 5 continued on next page*

*Figure 5 continued*

2200 (blue; **T–Y**). In panels M–R, mCitrine fluorescence is shown with and without merging with the propidium iodide stain image. Transgenic lines and genetic backgrounds used: (**A, G, M, S, T**) *tir1-10 TIR1-mCitrine#2*; (**B, H, N, S, U**) *afb1-3 AFB1-mCitrine#7*; (**C, O, V**) *afb2-3 AFB2-mCitrine#3*; (**I, S**) *afb2-3 AFB2-mCitrine#5*; (**D, J, P, S, W**) *afb3-4 AFB3-mCitrine#1*; (**E, K, Q, S, X**) *afb4-8 AFB4-mCitrine#3*; (**F, L, Y**) *afb5-5 AFB5-mCitrine#19* and (**R, S**) *afb5-5 AFB5-mCitrine#23*. Scale bars equal 25 µm (A–F), 50 µm (G–L), and 10 µm (M–R, T–Y).
The online version of this article includes the following source data and figure supplement(s) for figure 5:

**Figure supplement 1.** Complementation of mutant phenotypes by TIR1/AFB-mCitrine transgenes.
**Figure supplement 1—source data 1.** Source data for AFB4-mCitrine complementation.
**Figure supplement 2.** Comparison of *TIR1/AFB-mCitrine* lines.
**Figure supplement 3.** AFB1-mCitrine expression is unchanged in F$_1$ hybrids used for signal quantification.

mCitrine while that of AFB4-mCitrine line #1 that complemented the *afb4* phenotype was barely detectable (*Figure 5K*; *Figure 5—figure supplement 1D*, *Figure 5—figure supplement 2I–L*). In embryos, TIR1-, AFB2-, AFB3-, and AFB5-mCitrine accumulate fairly uniformly throughout the embryos and suspensors while AFB4-mCitrine's signal was close to background levels and AFB1-mCitrine was undetectable (*Figure 5T–Y*).

The subcellular localization of different TIR1/AFB proteins varied substantially (*Figure 5M–S*; *Figure 5—figure supplement 3*). We quantified this variation more precisely by measuring the relative level of each protein in the nucleus versus outside the nucleus in epidermal cells of the root elongation zone based on mCitrine fluorescence (*Figure 5S*). TIR1-mCitrine is primarily in nuclei while significant amounts of AFB2 through AFB5 are present in the cytoplasm. Strikingly, AFB1-mCitrine appears primarily outside the nuclei. The localizations are consistent across multiple lines and, when tested, different fluorescent protein tags (*Figure 5—figure supplement 2*). Hence, despite being from the same clade, TIR1 and AFB1 proteins exhibit contrasting patterns of primarily nuclear and cytoplasmic subcellular localizations, respectively.

## AFB1 plays a key role in the rapid auxin inhibition of root growth

Gravitropic curvature of the root is a rapid auxin-regulated growth response that requires asymmetric distribution of auxin between the upper and lower side of the root (*Sato et al., 2015*). According to the current model, auxin has two modes of action during gravitropism: a rapid nongenomic phase, followed by a transcriptional phase that is dependent on the TIR1/AFB proteins (*Shih et al., 2015*). Surprisingly, recent studies demonstrate that rapid, nongenomic auxin inhibition of root growth is dependent on the TIR1/AFBs (*Fendrych et al., 2018*). To determine the relative contribution of the TIR1/AFB family members to the rapid response, we measured the effect of 10 nM IAA on root growth in various *tir1/afb* lines over a 20 min time period (*Figure 6*; *Figure 6—figure supplement 1*; *Figure 6—videos 1–2*). The results in *Figure 6* show that each of the TIR1/AFBs contributes to rapid root growth inhibition but that, surprisingly, the *afb1-3* mutant is almost completely resistant to auxin indicating that AFB1 is the dominant auxin receptor for this response. Expression of *AFB1-mCitrine* under control of the *AFB1* promoter restored the wild-level of auxin response. The behavior of the *afb1* mutant is particularly remarkable since the mutant is not affected in any of the other auxin-regulated growth processes that we characterized, with the possible exception of lateral root formation. This includes long-term inhibition of root growth. Thus, the *afb1* mutant is a useful tool to discriminate between nongenomic and transcriptional auxin responses.

As a contrast we also examined the effect of auxin on etiolated hypocotyl growth in the mutant lines. This response is slower than the root response and depends on the canonical nuclear TIR1/AFB pathway (*Fendrych et al., 2016*). Dissected hypocotyl segments from etiolated seedlings were treated with 5 µM NAA and imaged every 10 min for 180 min. The response of the mutant lines was complex (*Figure 6—figure supplement 2*). Several lines were clearly resistant to auxin, particularly *tir1afb1245* and *tir1afb245*. Notably, comparison of these lines suggests that the *afb1* mutation did not contribute to resistance. Other lines also lacking both *AFB4* and *AFB5*—*afb1345*, *tir1345*, *afb1245*, and *afb12345*—display a moderate level of resistance. Finally, three lines, *tir1afb134*, *tir1afb23,* and *tir1afb124* are hypersensitive to auxin in this assay (*Figure 6—figure supplement 2*).

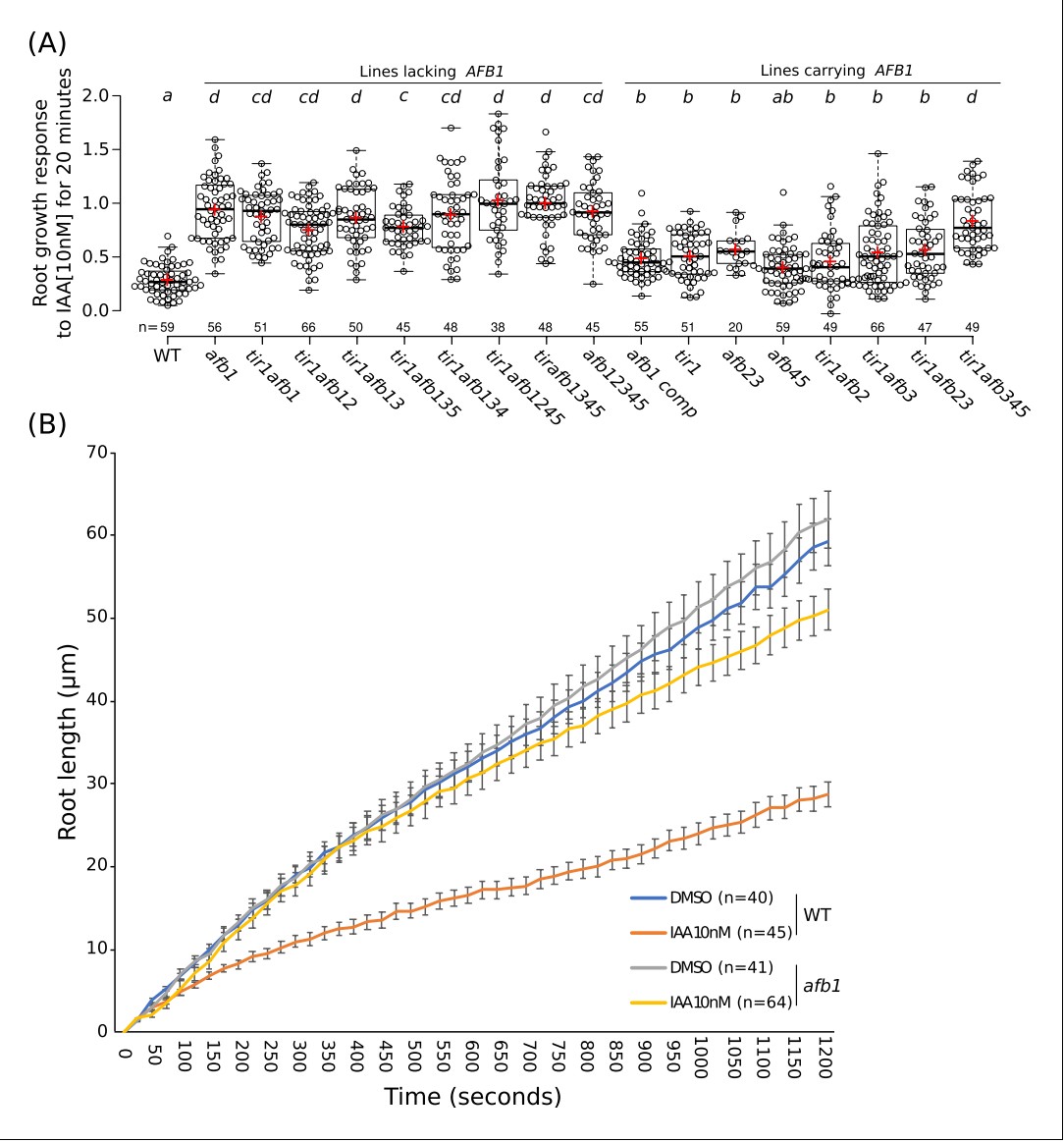

**Figure 6.** The role of *AFB1* in rapid inhibition of root elongation. (**A**) Plot of the root growth response of different genotypes to 10 nM IAA for 20 min. Black circles represent the response for one single root. Red crosses indicate the mean. Black bars indicate median. n indicates the number of roots obtained from three independent experiments. Letters indicate statistical differences according to one-way ANOVA coupled with post hoc Tukey honestly significant difference (HSD) test (p=0.05). (**B**) Graph of the root length in µm according to time in seconds of WT and *afb1* in DMSO and 10 nM IAA treatments (blue, gray, orange and yellow lines, respectively). Bars indicates standard deviation of the mean (SEM). n indicates the number of roots obtained from three independent experiments.

The online version of this article includes the following video, source data, and figure supplement(s) for figure 6:

**Source data 1.** Source data for root elongation assay.
**Figure supplement 1.** Time courses of root elongation.
**Figure supplement 2.** Graphs showing changes in length of hypocotyl segments treated with 5 µM NAA or 0.025% ethanol (Controls) for three hours.
**Figure supplement 2—source data 1.** Source data for hypocotyl elongation assay.
**Figure 6—video 1.** Movie of wild type root tip with mock (DMSO, left panel) and 10 nM IAA (right panel) treatments.
https://elifesciences.org/articles/54740#fig6video1
**Figure 6—video 2.** Movie of *afb1-3* root tip with mock (DMSO, left panel) and 10 nM IAA (right panel) treatments.
https://elifesciences.org/articles/54740#fig6video2

## AFB1 regulates the initial phase of root gravitropic response

Gravitropic root curvature is first apparent less than 10 min after a gravity stimulus (*Shih et al., 2015*). It has been proposed that the early stage of gravitropism is mediated by a non-genomic auxin response while prolonged root curvature requires auxin regulated transcription (*Sato et al., 2015*). Since AFB1, and to a lesser extent, the other TIR1/AFBs, contribute to nongenomic inhibition of root elongation, we wondered if they are required for the early gravitropic response. To test this possibility, we performed gravitropism assays on a number of *tir1/afb* lines (*Figure 7*; *Figure 7—figure supplement 1*). The gravitropic response can be divided into three phases. A slow or lag phase which occurs over the first 90 min in Col-0, followed by a 3 hr linear phase and finishing with a plateau phase. The *tir1afb345* line exhibits a slower gravitropic response during the linear phase and a reduced angle at plateau compared to WT (*Figure 7A*) as expected based on results with other *tir1/afb* mutant combinations (*Dharmasiri et al., 2005*). Strikingly, the *tir1afb1345* mutant exhibited an additional decrease in the gravitropic response during the initial lag phase demonstrating that AFB1 is required for this phase (*Figure 7A*; *Figure 7—figure supplement 1A*). The *tir1afb1* line showed a similar decrease in the lag phase compared to *tir1* (*Figure 7B*; *Figure 7—figure supplement 1B*). It seems likely that TIR1 also contributes to the early response since the *tir1afb1* double mutant showed a stronger delay compared to either single mutant. In addition, both *tir1* and *afb1* displayed a reduced early response in one of the two experiments (*Figure 7—figure supplement 1B*). Since other members of the family also confer low levels of auxin resistance in the rapid root growth response, these proteins may also make a small contribution to the early phase of gravitropism (*Figure 6A*). Further, both *afb1 AFB1-mCitrine* lines responded appreciably faster than the *afb1* mutant during the first 2 hr with the brighter of the two mCitrine lines, line #7, exhibiting a difference by 30 min (*Figure 7C*; *Figure 7—figure supplement 1C*). Interestingly, both mCitrine lines started to plateau earlier and at a reduced angle compared to wild type while *afb1* plateaued later and at an increased angle, suggesting that AFB1 and the rapid response also play a role at later stages of the gravitropic bending response.

## Discussion

The TIR1/AFB protein family has expanded through a series of gene duplication events that began before fern–seed-plant divergence. Despite the fact that three major subclades were established approximately 400 MYA (*Morris et al., 2018*), our genetic studies reveal that for most auxin-regulated growth processes, the TIR/AFB proteins retain largely overlapping functions (*Figure 8*). The striking exception to this general statement is the dominant role for AFB1 in rapid auxin inhibition of root growth. In general, *TIR1* is most important for normal growth and development, but *AFB5* and *AFB2*, and to a lesser extent *AFB3* and *AFB4*, also play significant roles. Spatial differences are also apparent; *TIR1* has a major role in the root while *AFB5* is relatively more important in hypocotyl and inflorescence development.

Although all six genes are broadly expressed, it appears that the relative importance of individual TIR1/AFB proteins in various organs are at least partly related to differences in expression. For example, *AFB5* is more broadly expressed than the other genes in the inflorescence while in the root, *TIR1* and *AFB2* are most highly expressed. The *AFB4* gene is expressed at a lower level in all tissues consistent with its relatively minor role. Additional differences in patterns of expression are also apparent, particularly in the inflorescence. Further studies will be required to determine if these differences are important.

Our studies demonstrate that the levels of the TIR1/AFB proteins are not uniform throughout the plant. This is true for individual members of the family and for total TIR1/AFB levels across different tissues and cell types. Earlier experiments also showed that TIR1/AFB levels can be dynamic in a changing environment (*Vidal et al., 2010*; *Wang et al., 2016*). These observations may have important implications for use of DII-Venus-based auxin sensors to estimate relative auxin levels, since levels of the sensor protein are dependent on both auxin and the TIR1/AFBs (*Brunoud et al., 2012*; *Liao et al., 2015*). Given the debate over an auxin-response asymmetry across shoot organ primordia (*Bhatia et al., 2019*; *Guan et al., 2019*), it is particularly interesting that we see an asymmetric distribution of TIR1-mCitrine across flower primordia.

It is important to emphasize that individual members of the family may have functions in particular environmental conditions. For example, the microRNA miR393 is known to target *TIR1, AFB2*, and

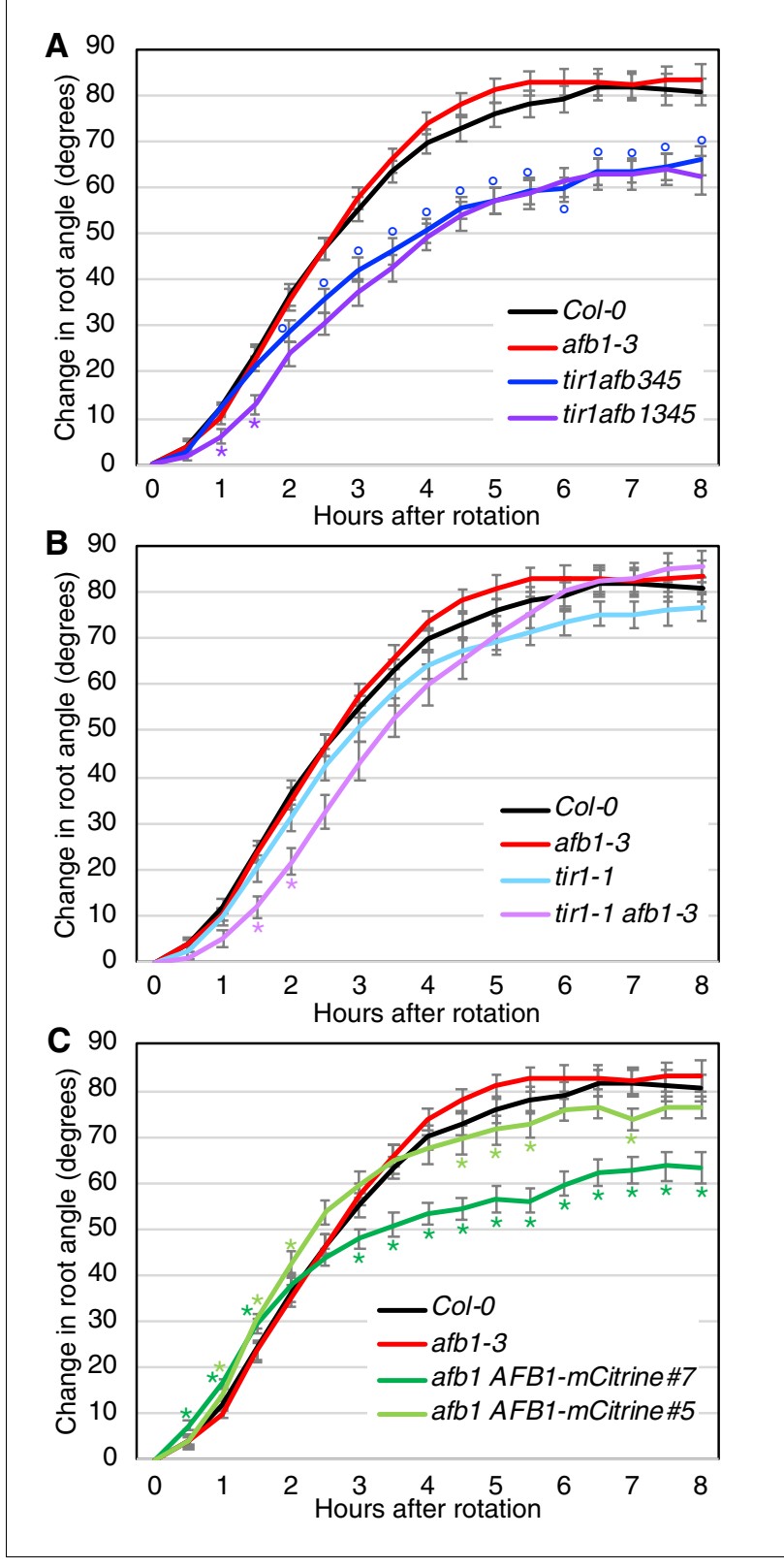

**Figure 7.** Gravitropic response of *tir1/afb* lines. Sixteen seedlings for each line were imaged every 30 min after rotating the plates 90° and the mean difference in the root-tip angle from the original angle ± SEM are plotted versus time. Col-0 and *afb1-3* are included in all panels for comparison. Time points at which lines differed from Col-0 are indicated by degree symbols (°) and differences between lines with and without the *afb1* mutation are

*Figure 7 continued on next page*

*Figure 7 continued*

indicated by asterisks (*) of the colors shown in the legend (*t*-test, p<0.05). Colors: black, Col-0; red, *afb1-3*; blue, *tir1afb345*; purple, *tir1afb1345*; cyan, *tir1-1*; lavender, *tir1-1 afb1-3*; light green, *afb1-3 AFB1-mCitrine#5*; and dark green, *afb1-3 AFB1-mCitrine#7*.

The online version of this article includes the following source data and figure supplement(s) for figure 7:

**Source data 1.** Source data for gravitropism assay.
**Figure supplement 1.** Gravitropic response of *tir1/afb* lines, repeat experiment.
**Figure supplement 1—source data 1.** Source data for gravitropism assay.

*AFB3* but not other members of the family (*Jones-Rhoades and Bartel, 2004*; *Navarro et al., 2006*). Regulation of miR393 abundance modulates the levels of these three TIR1/AFBs to facilitate various growth processes, such as lateral root formation and hypocotyl elongation in response to environmental signals (*Vidal et al., 2010*; *Pucciariello et al., 2018*).

Previous in vitro studies have documented some differences in the biochemical activity of members of the TIR1/AFB family (*Calderón Villalobos et al., 2012*; *Lee et al., 2014*). Similarly, an auxin-induced degradation assay in yeast reveals differences in the behavior of TIR1 and AFB2 (*Wright et al., 2017*). In contrast, our results do not reveal any biochemical specificity, except for *AFB1* (see below). Thus, a single *TIR1* or *AFB2* allele is sufficient to support viability throughout the plant life cycle albeit with dramatically reduced fertility. This contrasts to functional diversification seen in other well-studied gene families that diverged in a similar time frame such as the phytochrome photoreceptors and Class III HD-Zip transcriptional regulators (*Prigge et al., 2005*; *Franklin and Quail, 2010*; *Strasser et al., 2010*). It is possible that the retention of overlapping functions reflects stricter constraints on TIR1/AFB protein function. One possibility is that the different *TIR1/AFB* paralogs have been maintained because they contribute to the robustness of the auxin signaling system. Of course, specific functions may be revealed in future studies.

The importance of auxin in patterning of the developing embryo is well established (*Palovaara et al., 2016*). Auxin signaling, as evidenced by activity of the *DR5* reporter, is first apparent in the apical cell of the embryo (*Friml et al., 2003*). The essential role of auxin in the apical cell and later in the hypophysis is clearly demonstrated by the defects in the division of these cells in the *tir1afb235* quadruple and *tir1afb12345* sextuple mutant (*Figure 3*). Similar defects are observed in a number of other auxin mutants including those affecting response (*monopteros* and *bodenlos*), auxin synthesis (*yuc1 yuc4 yuc10 yuc11* and *taa1 tar1 tar2*) and transport (*pin1 pin3 pin4 pin7* and *aux1 lax1 lax2*) (*Berleth and Jürgens, 1993*; *Hardtke and Berleth, 1998*; *Hamann et al., 1999*; *Hamann et al., 2002*; *Friml et al., 2003*; *Cheng et al., 2007*; *Stepanova et al., 2008*; *Robert et al., 2015*). However, none of these lines exhibit the fully penetrant embryo-lethal phenotype observed for the *tir1afb235* quadruple and *tir1afb12345* sextuple mutants. In the other mutants, significant fractions of embryos escape embryo lethality and germinate, albeit often as rootless seedlings.

The expression of key embryonic markers in the mutants also reveals profound defects in embryonic patterning by the dermatogen stage. Although *tir1afb235* embryos form a morphologically normal hypophysis cell, this cell never expresses *NTT-YPet* or *WOX5:GFP*. The proliferation of suspensor cells in the mutant is associated with reduced expression of the suspensor marker *PIN7-Venus* and to a lesser extent *NTT-YPet* suggesting that the TIR1/AFB pathway is required to maintain the suspensor cell fate, consistent with an earlier study (*Rademacher et al., 2012*). *PIN1-Venus* is normally expressed in most cells distal from the hypophysis in globular embryos and in provascular tissue in later embryos, but it was expressed primarily in apical protodermal cells in *tir1afb235* mutants (*Figure 4M–N*). This is reminiscent of its pattern in the *monopteros* mutant, where some protodermal expression appears although the provascular expression is retained (*Breuninger et al., 2008*). Because both cotyledon specification and *PIN1* expression are influenced by auxin perception, it is unclear whether the mutant embryos lack the radial asymmetry that predicts cotyledon positioning or fail to elaborate on this asymmetry. It was surprising to observe that *PIN7-Venus* exhibits ectopic expression in the embryo proper. The reason for this is unclear but *PIN7* expression may normally be repressed in the embryo by a TIR1/AFB-dependent pathway. Given that PIN1 and PIN7 are normally expressed in non-overlapping domains in the embryo (*Friml et al., 2003*), one

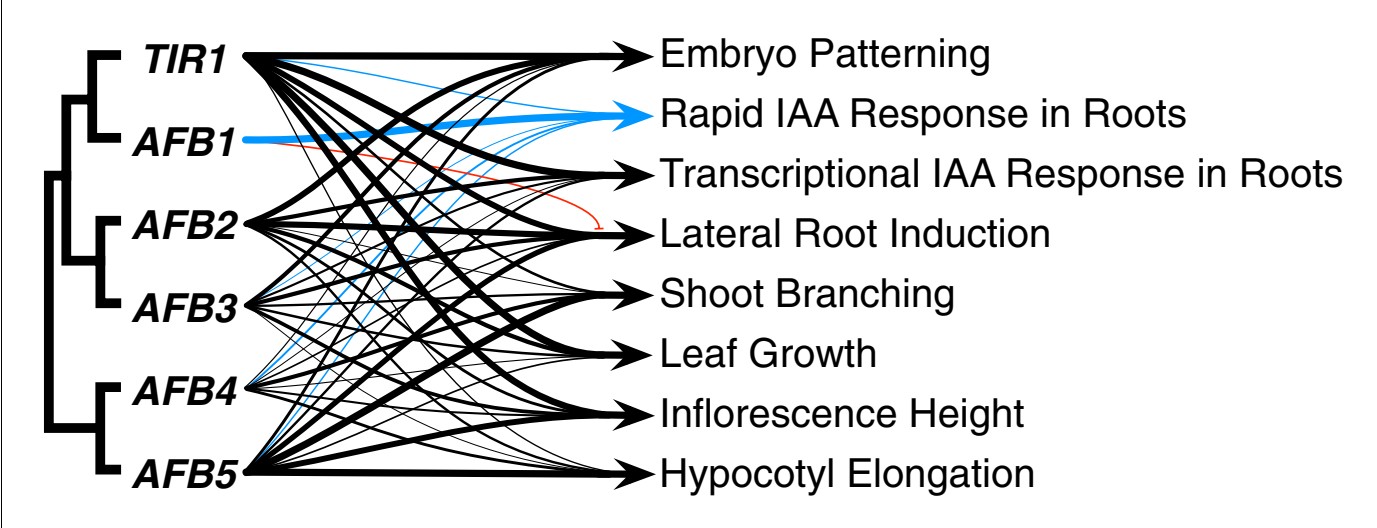

**Figure 8.** Summary of each *TIR1/AFB* gene's contributions to different responses. The line weights reflect the relative importance for each gene's roles. The blue lines represent contributions to the rapid IAA-mediated inhibition of root elongation and the red line with the bar end indicates the antagonistic role observed for AFB1 in lateral root production.

possibility is that the reduction in PIN1 expression in the mutants allows PIN7 to be expressed beyond its normal boundaries.

In contrast to the embryo, the role of auxin in gametophyte development is uncertain. Several reports suggest that auxin has an important role in patterning the female gametophyte (*Pagnussat et al., 2009*; *Panoli et al., 2015*; *Liu et al., 2018*). Others have argued against a role for auxin based on theoretical considerations as well as lack of evidence for an auxin response using several auxin reporters (*Lituiev et al., 2013*). Our studies suggest that the TIR1/AFB auxin receptors are not required for gametophyte development although we cannot rule out a minor role. It is important to note that we have not directly examined developing sextuple gametophytes and it is possible that there are minor defects that do not affect viability. We also can't eliminate the possibility of perdurance of TIR1/AFB proteins from the maternal tissue. Finally, it is possible that auxin is required, but acts through a non-canonical pathway such as that involving auxin binding to the ETTIN protein (*Simonini et al., 2016*).

*AFB1* is unique among the auxin co-receptors and appears to have undergone pronounced functional changes during the diversification of the Brassicales order since the *TIR1–AFB1* duplication in the At-β WGD around 80 to 90 million years ago (*Figure 1—figure supplement 1C*; *Edger et al., 2018*). Although AFB1 can interact with Aux/IAA proteins in an auxin-dependent manner, it does not appear to assemble into a Skp, Cullin, F-box containing (SCF) complex as efficiently as the other TIR1/AFBs and is not primarily localized to the nucleus where it could directly influence transcriptional responses (*Dharmasiri et al., 2005*; *Yu et al., 2015*; *Figure 5N and S*; *Figure 5—figure supplement 2C–D*). The F-Box substitutions in AFB1 affecting SCF assembly appeared between approximately 45 and 65 million years ago (*Figure 1—figure supplement 1C*; *Edger et al., 2018*). It is noteworthy that unlike the other *TIR1/AFB* genes that are broadly expressed in most cells, *AFB1* is expressed very highly in some tissues (root epidermis and vascular tissue) and not at all in others (meristematic pericycle and early embryos). Based on our genetic studies, AFB1 appears to have a negative effect on lateral root initiation in the *afb234* and *afb345* lines despite the fact that AFB1 is not expressed in the pericycle, the site of lateral root initiation, suggesting that this may be a non-cell-autonomous effect.

We find that AFB2 through AFB5 are distributed between the nucleus and the cytoplasm, at least in epidermal cells of the root (*Figure 5S*). In contrast, the paralogs TIR1 and AFB1 differ dramatically in being highly enriched in the nucleus and in the cytoplasm, respectively. In Arabidopsis roots, auxin

treatment results in very rapid responses including increased cytosolic Ca$^{++}$ levels, alkalinization of the apoplast and inhibition of root growth (*Shih et al., 2015*; *Dindas et al., 2018*; *Fendrych et al., 2018*). Because these events occurred too rapidly to involve transcription, it was assumed that they did not require the TIR1/AFB proteins. However, recent studies have demonstrated that two rapid responses, inhibition of root growth and membrane depolarization in root hairs, do require the TIR1/AFBs (*Dindas et al., 2018*; *Fendrych et al., 2018*). Surprisingly we find that the growth inhibition response is mediated primarily by AFB1. This may reflect the high level of AFB1 in the cytoplasm. It is not currently clear how deeply conserved the cytoplasmic localization of AFB1 is. It is possible that AFB1's specialization is a relatively recent event and that the responsibility of mediating the rapid response is shared by multiple TIR1/AFB proteins in other plant lineages. An answer to this question will require further information on the molecular basis for AFB1 localization.

It has been proposed that the rapid nongenomic auxin response in the Arabidopsis root has a role in early stages of root gravitropism (*Sato et al., 2015*), and our results support this idea. Although the *afb1* mutant has only a modest effect on gravitropism by itself, in combination with *tir1* or *tir1afb345,* it confers a clear decrease in early gravity response. It is surprising that the *afb1* mutation has only a modest effect on root gravitropism given the nearly complete absence of the rapid nongenomic auxin response. This may be a reflection of the gravitropic assay we have employed. Further detailed studies of the gravitropic response may reveal a more substantial role for the rapid response. The fact that two *AFB1-mCitrine* lines both appear to affect the early response as well as the angle at the plateau phase, hint at additional complexity. Although the rapid auxin response has only been described thus far in Arabidopsis, it is probably not unique to the Brassicales given that a relatively fast gravitropic response is common in diverse seed plants (*Zhang et al., 2019*). If the rapid auxin response evolved prior to the TIR1–AFB1 duplication event and the ancestral TIR1/AFBs contributed to both the nuclear genomic and cytoplasmic nongenomic auxin responses, the differences between TIR1 and AFB1 represent an elegant example of subfunctionalization of AFB1 to a role in the non-genomic response and, possibly, of TIR1 to specialize in the nuclear auxin response. Furthermore, as AFB1 has a major role in the rapid response but little or no function in the transcriptional response, the *afb1* mutant provides a useful tool to separate the two responses.

## Materials and methods

### Phylogeny

The sources for the amino-acid sequences (*Figure 1—figure supplement 1A*) and CDS (*Figure 1—figure supplement 1C*) are listed in *Supplementary file 3* (*Jiao et al., 2011*; *Goodstein et al., 2012*; *Johnson et al., 2012*; *Matasci et al., 2014*; *Wickett et al., 2014*; *Xie et al., 2014*; *One Thousand Plant Transcriptomes Initiative, 2019*). Taxa were selected based on availability, quality, and diverse sampling at key nodes. A reduced set was included for COI1 homologs. The *AFB1* genes from *Camelina hispida*, *C. laxa*, and *C. rumelica* were amplified from genomic DNA using Phusion Polymerase (New England Biolabs or ThermoFisher) and primers to regions of the 5′ and 3′ UTRs conserved in all three *C. sativa AFB1* genes in the *C. sativa* genome (*Kagale et al., 2014*). The PCR products were subcloned, and three *C. hispida* and *C. laxa* clones and a single *C. rumelica* clone were sequenced. The *CamhiAFB1* and *CamlaAFB1* sequences included in analysis appeared in two of the three clones (GenBank accession numbers MK423960–MK423962).

To build the alignment of F-Box-LRR protein sequences, sequences from distinct subclades were aligned using T-COFFEE v11.00 (*Notredame et al., 2000*) to identify and trim unique unalignable regions from individual sequences before aligning the whole set. Ambiguous regions of the full alignments were removed in Mesquite v3.5 (*Maddison and Maddison, 2018*). The raw alignment of nucleotide CDS sequences of Brassicales *TIR1/AFB1* genes was adjusted so that gaps fell between adjacent codons. Phylogenetic trees were inferred using MrBayes v3.2.6 (*Ronquist et al., 2012*). For the TIR1/AFB/XFB/COI1 phylogeny, a total of six runs of four chains were split between two Apple iMac computers using the parameters aamodelpr = mixed, nst = 6, and rates = invgamma. Only four of the six runs had converged after 16 million generations, so the analysis was restarted with three runs each starting with the best tree from one of the initial runs and with more heating (temp = 0.5) for 10 million generations. The *TIR1/AFB1* nucleotide alignments were partitioned by codon position

with ratepr = variable, nst = 6, rates = invgamma with three runs of 4 chains run for five million generations. The consensus trees were viewed using FigTree v1.4.4 (*Rambaut, 2018*).

## Mutants

The alleles used—*tir1-1, tir1-9, tir1-10, afb1-3, afb2-1, afb2-3, afb3-1, afb3-4, afb4-8,* and *afb5-5*— have been described previously (*Ruegger et al., 1998*; *Dharmasiri et al., 2005*; *Parry et al., 2009*; *Prigge et al., 2016*). Seeds from *Camelina* species were provided by the United States National Plant Germplasm System (USDA-ARS, USA): *C. hispida* (PI 650133), *C. laxa* (PI 633185), and *C. rumelica* (PI 650138). Unless noted, plants were grown at 22°C long-day (16:8) conditions on ½×Murashige and Skoog media with 0.8% agar, 1% sucrose, and 2.5 mM MES, pH 5.7, or in a 2:1 mixture of soil mix (Sunshine LC1 or ProMix BX) and vermiculite. Leaf DNA was isolated with a protocol adapted from *Edwards et al. (1991)* to use steel BBs (Daisy Outdoor Products), 2 ml microcentrifuge tubes, and 20-tube holders (H37080-0020, Bel-Art). See *Supplementary file 4* for primers used for genotyping.

## Fluorescent marker lines were described previously:

*NTT-2×YPet* (*Crawford et al., 2015*), *PIN7-GFP* (*Blilou et al., 2005*), *DR5rev:3×Venus*-N7 (*Heisler et al., 2005*), *WOX5:GFPER* (*Blilou et al., 2005*). The recombineered *PIN1-Venus* and *PIN7-Venus* markers (*Zhou et al., 2011*) were obtained from the Arabidopsis Biological Resource Center (CS67184 and CS67186). Previously characterized *PIN1-GFP* lines could not be used because of tight linkage to *AFB2* (CS9362) and co-segregation with Ler-derived enhancers of the *afb2/+ tir1afb35* phenotype (CS23889). Each marker was introgressed into lines segregating the *tir1afb235* quadruple mutant by two sequential crosses, PCR genotyping, and selfing. Marker line homozygosity was confirmed in F$_1$ seedlings from test crosses to WT. The *UBQ10:H2B-mTurquoise2* marker was assembled by combining the pK7m34GW destination vector (*Karimi et al., 2007*), UBQ10prom_P4P1R (*Jaillais et al., 2011*) (provided by Nottingham Arabidopsis Stock Centre, N2106315), H2B_noStop/ pDONR207 (provided by Frederic Berger), and 2×mTurqoise2/pDONR-P2RP3 using the LR Recombinase System (Life Technologies). For 2×mTurqoise2/pDONR-P2RP3, the mTurquoise2 coding sequence (*Goedhart et al., 2012*); provided by Joachim Goedhart) was amplified using primers mTU2_P2RP3_F and mTU2_P2RP3_wSTOP_R primers and recombined into pDONR-P2RP3 vector in a BP reaction to give mTURQUOISE2/pDONR-P2RP3. This plasmid was amplified using the INS_mTU2_P2RP3_F and INS_mTU2_P2RP3_wSTOP_R primers and the BB_mTU2_ P2RP3_F and BB_mTU2_ P2RP3_R primers. The PCR products were assembled by Gibson cloning (New England Biolabs) to give 2×mTurqoise2/pDONR-P2RP3. The *UBQ10:H2B-mTurquoise2* transgene was introduced to Col-0 plants as described (*Simon et al., 2014*).

## Fluorescently tagged TIR1/AFB lines

Genomic regions containing each of the *TIR1/AFB* genes were amplified using Phusion polymerase (New England Biolabs or ThermoFisher) from corresponding genomic clones (JAtY51F08, JAtY62P14, JAtY53F15, JAtY61O12, and JAtY52F19) except for *AFB3* which was amplified from Col-0 genomic DNA. See *Supplementary file 4* for primers used. The PCR products were cloned into pMiniT (New England Biolabs), and the stop codon was altered to create a *Nhe*I site using site-directed mutagenesis. An *Xba*I fragment containing either mCitrine (*Griesbeck et al., 2001*), mOrange2 (*Shaner et al., 2008*), or mCherry (*Shaner et al., 2004*) preceded by a short linker (either Arg-Gly$_5$-Ala or Arg-Gly$_4$-Ala) was ligated into the *Nhe*I sites. The genomic regions including the fluorescent protein genes were inserted in the *Mlu*I site of pMP535 (*Prigge et al., 2005*) as *Asc*I fragments (*AFB5*) or as *Mlu*I-*Asc*I fragments (others). To produce the sextuple-complementation construct, the *TIR1-mOrange2* fragment was cloned into pMP535 as above, then *AFB2-mCitrine* was inserted into the re-created *Mlu*I site followed by *AFB5-mCherry* into its re-created *Mlu*I site. The constructs were introduced into the following strains by floral dip (*Clough and Bent, 1998*): *tir1/+ afb5/+ afb1234* progeny (sextuple-complementation construct), *tir1afb23* (TIR1-mCitrine, AFB3-mCitrine, and AFB3-mEGFP), *tir1afb1245* (AFB2-mCitrine), *afb45* (AFB4-mCitrine and AFB4-tdTomato), *afb5-5* (AFB5-mCitrine, and *afb1-3* (AFB1-mCitrine). Basta-resistant candidate lines were selected based on complementation of visible phenotypes (except for *AFB1-mCitrine*) then crossed to get them into the appropriate mutant backgrounds. Once in the sextuple-mutant background, the

complementation transgene was maintained as a hemizygote by checking siliques for aberrant embryos or aborted seeds. The *afb5-5 AFB5-mCitrine #9* and *#19* lines were described previously (*Prigge et al., 2016*).

## Microscopy

For confocal microscopy of the root meristem, five- to seven-day-old seedlings were stained in a 10 µg/ml aqueous solution of propidium iodide for one minute, rinsed in water, mounted with water, and viewed with either a Zeiss LSM 880 inverted microscope or a Zeiss LSM 710 inverted microscope. Embryos were fixed and stained with SCRI Renaissance 2200 (SR2200; Renaissance Chemicals, UK; *Crawford et al., 2015*). Briefly, using fine forceps and a 27-gauge needle as a scalpel, developing seeds were dissected from siliques and immediately immersed in fix solution (1 × PBS, 4% formaldehyde (Electron Microscopy Sciences, 15713), and 0.4% dimethyl sulfoxide) in a six-well plate with 100µ-mesh strainers. A vacuum was pulled and held three times for 12 min each time, before rinsing twice with 1 × PBS for 5 min. The embryos were transferred to SR2200 stain [3% sucrose, 4% diethylene glycol, 4% dimethyl sulfoxide and 1% SR2200 and stained overnight with vacuum pulled and released 3–4 times. Seeds were mounted (20% glycerol, 0.1 × PBS, 0.1% dimethyl sulfoxide, 0.1% SR2200, and 0.01% Triton X100) and the embryos were liberated by pressing on the coverslip. To detect mCitrine in the shoot apices, we removed stage 5 and older floral buds using fine forceps, fixed and rinsed (as with the embryos), soaked in ClearSee (*Kurihara et al., 2015*) for seven to ten days changing the solution every two to three days, and then stained with basic fuchsin (not shown) and Fluorescent Brightener 28 (Calcofluor White M2R) as described (*Ursache et al., 2018*). Confocal image channels were merged using ImageJ or FIJI (*Schindelin et al., 2012*; *Schneider et al., 2012*). Cleared embryos were viewed by mounting dissected ovules in a solution containing 2.5 g chloral hydrate dissolved in 1 ml 30% glycerol and viewed with a Nikon E600 microscope.

## Fluorescence quantification

In order to infer the amounts of TIR1/AFB protein inside and outside the nucleus, 40 × magnification images of epidermal cells in the elongation zone of each TIR1/AFB-mCitrine lines were captured. Because the nuclei of AFB1-mCitrine-expressing cells are not apparent, the $F_1$ of a cross with a plant with a *UBQ10:H2B-mTurquoise2* transgene was used to delineate the nucleus. Using FIJI (*Schindelin et al., 2012*), regions of interests including the entire cell (cell, based on propidium iodide staining), the nucleus (nuc, based on mCitrine or mTurquoise2 signal), and a cell-sized region outside the root (bg, background) were drawn using the freehand selections tool, and the area and mean gray values were measured for the mCitrine channel for each. The percent nuclear was calculated using the equation $\%_{nuc} = [\text{Area}_{nuc} \times (\text{Mean}_{nuc} - \text{Mean}_{bg})] \div [\text{Area}_{cell} \times (\text{Mean}_{cell} - \text{Mean}_{bg})]$.

## Phenotype comparisons

The viable *tir1afb* lines were divided based on whether they contained the *tir1* mutation, and the two batches were grown sequentially. The *afb123* line included in the initial batch displayed a long-hypocotyl phenotype that may have been picked up after an earlier cross to the *afb4-2* mutant, so a third batch was made up of alternative isolates for five lines whose pedigrees included a cross to *afb4-2*. Each batch included Col-0 and *tir1-1*. Seeds were surfaced sterilized, stratified in water for five days, spotted onto ½ Murashige and Skoog (MS) medium containing 1% sucrose, and incubated in a light chamber (22°C). Twelve five-day-old seedlings for each genotype were transferred to 120 mm square plates containing the same medium containing either 0, 20, or 100 nM IAA (batch a), 0, 100, or 500 nM IAA (batch b), or 0, 20, 100, or 500 nM IAA (batch c). Each plate received six seedlings from six genotypes spread out over two rows. Seedlings for each genotype were present on the top row of one plate and the lower row on a second plate placed in a different part of the growth chamber after marking the position of the root tips with a marker and scanning with Epson V600 flatbed scanners. The plates were scanned again after 72 hr (96 hr for batch c), and the growth was measured using imageJ. The plates containing 100 nM IAA were grown for a fourth day before the numbers of lateral roots protruding through the epidermis were counted using a dissecting microscope. Five seedlings from the no-IAA control plates were transferred to soil in 6 cm pots and grown an additional 34 days. The genotypes for two plants per line were confirmed by PCR. For

each 42-day-old plant, the height from the rosette to the tip of the longest inflorescence and the maximum rosette diameter were measured, and the numbers of branches of at least 1 cm were counted. The IAA effects on root elongation data is presented as the percent relative to the growth without IAA ± the relative standard error of the ratio. For the gene effect analyses, the averages from each batch were normalized using measurements for Col-0 and *tir1-1* plants that were included with each batch.

## Time lapse imaging of root growth

Seeds were sown on ½ MS medium containing 1% sucrose and 0.8% agar and stratified for 2–3 days at 4°C. Approximately fifteen 5-day-old seedlings were transferred to culture chambers (Lab-Tek, Chambers #1.0 Borosilicate Coverglass System, catalog number: 155361) containing the same agar medium supplemented with DMSO or IAA 10 nM (stock solution at 10 µM in DMSO). The transfer of seedlings was completed within 45–60 s. Images were acquired every 25 s for 20 min representing 50 images per root using Keyence microscope model BZ-X810 with 4 × lens.

Images obtained for one field were stacked and cropped to the region of interest (ROI). An auto threshold using the method 'Default' was applied. In addition, the 'erode', 'despeckle' and 'Remove outliers' (radius 10, threshold 50) functions were used to smooth the image and remove the remaining background. Each root tip was selected and the 'Feret Distance' within the ROI (which corresponds to the longest distance in an object) was determined for each root. Image processing was automated with an ImageJ macro, *Supplementary file 8*. For each time point the 'Feret Distance' root growth was calculated by subtracting the initial 'Feret Distance'. The values obtained were used to generate graphs. For each genotype, the experiment was repeated three independent times.

To determine the effect of auxin on root growth throughout the experiment, the area under each curve of auxin-treated roots was determined and divided by the corresponding value for roots grown on DMSO condition to calculate the root growth response to IAA. A response value of 1 indicates that IAA had no effect on root growth. The effect of IAA on root growth was determined this way to account for differences in root growth between genotypes on DMSO.

Each sample was subjected to four different normality tests (Jarque-Bera, Lilliefors, Anderson-Darling and Shapiro-Wilk). Samples were considered as a Gaussian distribution when at least one test was significant ($p=0.05$). As a normal distribution was observed a one-way ANOVA coupled with a post hoc Tukey Honestly Significant Difference test was performed ($p=0.05$).

## Hypocotyl segment elongation assay

Measurements of etiolated hypocotyl elongation were carried out essentially as described previously (*Fendrych et al., 2016*; *Li et al., 2018*). Seeds were sterilized and stratified for four (set 1) or five (set 2) days before plating. After 6 hr of light treatment, the plates were wrapped in aluminum foil and sealed in a cardboard box for 66 hr. The plates were opened in a room lit only with an LED desk lamp with six layers of green cello film (Hygloss Products) filtering the light. Using a dissecting microscope with its light source filtered with six sheets of green cello film, the roots and cotyledons were excised using razor blades and the hypocotyls transferred to plates containing depletion medium (DM: 10 mM KCl, 1 mM MES pH 6, 1.5% phytagel) overlain with a piece of cellophane (PaperMart. com). After 30 to 80 min on DM, the hypocotyl segments were transferred to treatment plates (DM plus either 5 µM NAA or the equivalent amount of solvent (0.025% ethanol). Eight to sixteen hypocotyls were transferred for each genotype and treatment except for there being only five control-treated *tir1afb23*. Using Epson V600 flat-bed scanners, the plates were scanned at 1200 dpi 30–60 s after transfer then every ten minutes for three hours. The segments were measured using a FIJI macro that applied 'Auto Threshold' (Default), 'Despeckle,' 'Remove Outliers' (radius = 2 threshold = 50 which = Bright), then returned the 'Feret Distance' for each. For each segment at each time point, the Feret distance was subtracted from the initial Feret distance. The lengths were converted to µm using the conversion 21.16667 µm/pixel. In the second experiment, Col-0 hypocotyls were dissected first and a second batch was dissected after the other genotypes to test whether the length of time on DM affected the assay. Measurements for the two batches were only different at the 20 min time point ($p<0.05$ in two-tailed *t*-test).

## Gravitropism assay

In the experiments corresponding to *Figure 7*, six-day-old seedlings were positioned on four 120 mm square plates such that four seedlings of each genotype were in different positions in the four plates to reduce position effects. The plates were placed in the growth chamber vertically for an hour, scanned with an Epson V600 flat-bed scanners, then returned to the chamber vertically but rotated 90° from the original orientation. Plates were re-scanned every 30 min for 8 hr. For each root tip at each time point, the angle was measured in FIJI by drawing a line drawn from the medial point two-root-widths from the root tip to the root tip. These angles were corrected for scan-to-scan differences in plate orientation by measuring the angle of a horizontal line on the plate in each image. The mean changes in root-tip angle from that at time zero ± S.E.M. for each genotype at each timepoint was plotted. The experiments corresponding to *Figure 7—figure supplement 1* were carried out in the same manner except that each plate contained a single genotype, and the seedlings were not repositioned onto different plates prior to rotation and scanning.

## Acknowledgements

We thank Yingluo Wang and Diane Le for technical assistance, Brian Crawford for help with embryo microscopy, and the Arabidopsis Biological Resource Center and the US National Plant Germplasm System for seeds. This work was supported by a grant from the NIH (GM43644 to ME) and by start-up funds from the Salk Institute of Biological Studies (WB). MP was supported by a long-term post-doctoral fellowship (LT000340/2019 L) by the Human Frontier Science Program Organization, and NK was supported in part through a UC San Diego Biological Sciences Eureka! Summer Research Scholarship. RB was supported by BBSRC Discovery and Future Food Beacon Nottingham Research Fellowships.

## Additional information

### Funding

| Funder | Grant reference number | Author |
|---|---|---|
| National Institutes of Health | GM43644 | Mark Estelle |
| Human Frontier Science Program | LT000340/2019-L | Matthieu Platre |
| Biotechnology and Biological Sciences Research Council | Research fellowship | Rahul Arvind Bhosale |

The funders had no role in study design, data collection and interpretation, or the decision to submit the work for publication.

### Author contributions

Michael J Prigge, Conceptualization, Formal analysis, Supervision, Validation, Investigation, Visualization, Methodology, Writing - original draft, Writing - review and editing; Matthieu Platre, Conceptualization, Formal analysis, Validation, Investigation, Visualization, Methodology, Writing - original draft, Writing - review and editing; Nikita Kadakia, Data curation, Investigation, Methodology; Yi Zhang, Investigation, Methodology; Kathleen Greenham, Resources; Whitnie Szutu, Bipin Kumar Pandey, Rahul Arvind Bhosale, Resources, Investigation; Malcolm J Bennett, Supervision, Funding acquisition, Writing - review and editing; Wolfgang Busch, Resources, Supervision; Mark Estelle, Conceptualization, Resources, Formal analysis, Supervision, Funding acquisition, Writing - original draft, Project administration, Writing - review and editing

### Author ORCIDs

Michael J Prigge (iD) https://orcid.org/0000-0003-0671-2538
Kathleen Greenham (iD) http://orcid.org/0000-0001-7681-5263
Whitnie Szutu (iD) https://orcid.org/0000-0002-2083-7241
Bipin Kumar Pandey (iD) http://orcid.org/0000-0002-9614-1347

Rahul Arvind Bhosale (iD) http://orcid.org/0000-0001-6515-4922
Mark Estelle (iD) https://orcid.org/0000-0002-2613-8652

**Decision letter and Author response**
Decision letter https://doi.org/10.7554/eLife.54740.sa1
Author response https://doi.org/10.7554/eLife.54740.sa2

## Additional files

### Supplementary files
- Supplementary file 1. Summary of phenotypes for mutant combinations.
- Supplementary file 2. Transmission of the sextuple mutant through megagametophytes and pollen.
- Supplementary file 3. List of databases for the sequences used in making the gene trees.
- Supplementary file 4. List of primers used for cloning and genotyping.
- Supplementary file 5. Nexus file for inferring the F-Box-LRR family tree.
- Supplementary file 6. Nexus file for inferring the TIR1+AFB1 tree.
- Supplementary file 7. Statistical tests.
- Supplementary file 8. Time Lapse Analysis 20 min Macro.
- Transparent reporting form

### Data availability
All data generated or analysed during this study are included in the manuscript and supporting files. Source data files have been provided for Figure 1—figure supplement 2B, Figure 1—figure supplement 3, Figure 1—figure supplement 4, Figure 5—figure supplement 1, Figure 6, Figure 6—figure supplement 2, Figure 7, Figure 7—figure supplement 1.

The following previously published dataset was used:

| Author(s) | Year | Dataset title | Dataset URL | Database and Identifier |
|---|---|---|---|---|
| Leebens-Mack JH, Wong GK, One Thousand Plant Transcriptomes Initiative | 2019 | Data packages for One Thousand Plant transcriptomes and phylogenomics of green plants | http://doi.org/10.25739/8m7t-4e85 | Data Commons, 10.25739/8m7t-4e85 |

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
