## [Decision Letter]

**Acceptance summary:**

This study genetically dissects TIR1/AFB-dependent auxin perception mechanisms and we are very pleased to cover the authors’ extensive work at *eLife*. Moreover, the authors provide valuable research tools, which will be truly of great interest to various plant research fields.

**Decision letter after peer review:**

[Editors’ note: the authors submitted for reconsideration following the decision after peer review. What follows is the decision letter after the first round of review.]

Thank you for submitting your work entitled "The *Arabidopsis* TIR1/AFB auxin receptors are essential early in embryogenesis and have broadly overlapping functions" for consideration by *eLife*. Your article has been reviewed by three peer reviewers, one of whom is a member of our Board of Reviewing Editors, and the evaluation has been overseen by a Senior Editor.

Our decision has been reached after consultation between the reviewers. Based on these discussions and the individual reviews below, we regret to inform you that the current version of your manuscript was rejected at this stage.

The reviewers were largely positive about your work, but also pointed out some shortcomings that precludes the publication of your manuscript in the present form. It is policy of *eLife* not to hold back authors in a lengthy reviewing process. However, in this case we also discussed at the editorial level and would like to encourage you to resubmit to *eLife* if you can satisfactorily address the main concerns of our expert reviewers. A major concern of the three reviewers relates to the description of the early embryo phenotypes, which would require additional quantitative assessments. Moreover, the reviewers were wondering if you could use your genetic resource to assess eminent questions, such as whether the TIR1/AFBs have diversified roles in slow and fast auxin responses.

*Reviewer #1:*

In this work the Estelle lab dissects the functional redundancy of TIR/AFBs auxin receptors. This work confirms numerous previous assumptions and reveals that TIR-dependent perception of auxin is indeed required for embryogenesis in *Arabidopsis*.

The authors provide an impressive list of mutant combinations, which also allowed them to quantitatively assess the contribution of TIR and AFB proteins to certain developmental aspects. This will be a great resource for the auxin field, but as some of the mutant combinations have been already published, the postembryonic roles of TIR/AFBs have been discussed in previous publications. Accordingly, it would be really nice if the authors could still go beyond this. Considering that TIR-dependent perception is involved in slow (hypocotyl) and fast (root) auxin responses. I am wondering if the authors could use this genetic resource to dissect whether the importance of distinct TIR/AFBs deviate in these slow and fast responses.

The authors report certain differences in expression and subcellular localization of TIR and AFB proteins. Especially the latter claim needs higher resolution (cellular close ups) and quantification (ratio of nuclear and cytosolic signal). Does the subcellular localization of TIR and AFB proteins change in response to auxin? Besides, it is important to mention/reveal that all these markers are functional.

*Reviewer #2:*

The manuscript by Prigge et al., entitled "The *Arabidopsis* TIR1/AFB auxin receptor genes are essential early in embryogenesis and have broadly overlapping functions", presents an extensive phenotypic analysis of the sextuple t*ir1afb12345* and related mutant combinations. The manuscript starts with a phylogenetic analysis of the TIR1/AFB gene family. It then describes the vegetative and reproductive phenotypes of various mutant combinations (upon to the embryo-lethal sextuple mutant). It demonstrates the importance of auxin signalling for early embryo development and point out the specific function of each receptor protein.

The manuscript is well written and the methods are well described.

I am impressed by the amount of work that was necessary to generate the genetic material studied in this work (63 allele combinations), as well as by the extensive phenotyping analysis of this material. Details of the measurements are provided, except for the embryo phenotype quantification.

In the subsection “Early-Embryo Defects of the *tir1afb235* and *tir1afb12345* mutant lines” when the authors detail the observed segregation and types of defects observed during embryo development, they are rather unprecise. There are no exact number or percentage of observed defects in n embryos of what genotype/s. However, this is part is an essential part of the manuscript (and in the title). Could the authors provide quantitative data for: "roughly one quarter of the embryos from each line lacked cotyledons…", for "a rate close to the expected 1/16 ratio" and "expected number of progeny"?

If I am correct (from data provided in Figure 1—figure supplement 2), TIR1 and AFB2 genes sit on the same chromosome. Therefore, the phenotype segregation on *tir1*/+ *afb2*/+ *afb345* might be deviant from the classical Mendelian segregation for 2 alleles, as TIR1 and AFB2 loci may be linked. This is relevant because the embryo phenotypes are dependent on these 2 alleles. This concerns data on Figure 3 and Figure 3—figure supplement 2 (transmission of the sextuple mutant pollen). In the latter one, could you provide the number of crosses analysed per genotype? Some n are rather low (13, 19, 22), which may affect the allelic segregation per genotype. It should be clarified that you are looking at the segregation of the sextuple homozygous offspring versus the rest. If I am not mistaken those sextuple embryos are lethal and this can then affect the segregation analysis. Can you please indicate what tissue was genotyped in this analysis?

One last point concerns the discussion on the expression pattern changes of marker genes in the mutant embryos. I was wondering if there is any information about the expression dependency of those markers on auxin. Is the expression of those genes, auxin-responsive? This is, for example, the case for PIN1, a target of MP (Schlereth et al., 2010; Robert et al., 2015). This would in part explain their absence of expression in the *tir1afb235* mutant embryos. PIN1 is absent in the inner cells in the mutant embryos, very similar to what was observed in *taa1 tar1 tar2* and *yuc 1 yuc 4 yuc 10 yuc 11* embryos (Robert et al., 2013) when auxin is not provided to the MP-dependent auxin signalling pathway. As for PIN7 switching to the embryo versus the suspensor, this might be the consequence of PIN1 absence and genetic redundancy: PIN7 would take over PIN1 expression domain as it was observed for PIN4 in *pin7* embryos (Vieten et al., 2005). At least, this discussion is also missing a conclusive remark.

*Reviewer #3:*

In this paper, the authors have analysed a large number of mutant combination in order to reveal the contribution of TIR/AFB auxin receptors to plant development. They have focused on phenotypes of the rosette and the inflorescence, seed number, root growth, and early embryos. The major outcome is that TIR/AFB auxin receptors have largely overlapping functions. Although I feel sympathetic for the efforts the authors have undertaken (keeping track of so many mutant combination itself is worthwhile to note), unfortunately little novel mechanistic insight has been obtained. The authors provide interesting speculations in the Discussion part, such as whether AFB1 might sequester auxin, but these ideas are not experimentally addressed. Altogether, I think this paper provides an important reference point for future work in the auxin field, but at this stage I am not sure whether it contains sufficient novel insight for a broader audience.

---

## [Author Response]

[Editors’ note: the authors resubmitted a revised version of the paper for consideration. What follows is the authors’ response to the first round of review.]

Reviewer #1:In this work the Estelle lab dissects the functional redundancy of TIR/AFBs auxin receptors. This work confirms numerous previous assumptions and reveals that TIR-dependent perception of auxin is indeed required for embryogenesis in Arabidopsis.The authors provide an impressive list of mutant combinations, which also allowed them to quantitatively assess the contribution of TIR and AFB proteins to certain developmental aspects. This will be a great resource for the auxin field, but as some of the mutant combinations have been already published, the postembryonic roles of TIR/AFBs have been discussed in previous publications. Accordingly, it would be really nice if the authors could still go beyond this. Considering that TIR-dependent perception is involved in slow (hypocotyl) and fast (root) auxin responses. I am wondering if the authors could use this genetic resource to dissect whether the importance of distinct TIR/AFBs deviate in these slow and fast responses.

We have now performed these experiments, looking at both rapid inhibition of root elongation and slower transcription-dependent stimulation of hypocotyl elongation. These results are described in Figure 6 and Figure 6—figure supplements 1-4. We were very surprised to learn that AFB1 makes the largest contribution to the rapid response in the root. The implications of this result are described in the text. In contrast, all members of the family except AFB1 contribute to hypocotyl elongation. Some genotypes are hypersensitive to auxin. This result is difficult to interpret but may be related to mis-regulation of Aux/IAA genes, themselves targets of the pathway, resulting in a net decrease in Aux/IAA protein level. In any case this result is clear indication of the complexity of the auxin signaling network.

Since the rapid response is thought to have an important role in root gravitropism, we examined the gravitropic response in several *tir1/afb* lines. These results are presented in Figure 7 and Figure 7—figure supplement 1. Our results clearly indicate that AFB1 contributes to the early phase of root gravitropism. This is the first clear demonstration of a physiological role for a rapid auxin response.

The authors report certain differences in expression and subcellular localization of TIR and AFB proteins. Especially the latter claim needs higher resolution (cellular close ups) and quantification (ratio of nuclear and cytosolic signal). Does the subcellular localization of TIR and AFB proteins change in response to auxin? Besides, it is important to mention/reveal that all these markers are functional.

We included quantification and higher magnification images of the cellular localizations in Figure 5 and included additional non-merged images in Figure 5—figure supplement 3.

We can confirm that each of the fusion proteins rescues the mutant phenotype. Except for AFB1’s, each mCitrine transgene was transformed into multiply mutant plants with obvious phenotypes – *tir1afb23* (TIR1, AFB3), *tir1afb1245* (AFB2), and *afb45* (AFB4) – and selected lines that suppressed the phenotypes. We added a figure (Figure 5—figure supplement 1) showing the inflorescence phenotypes of the TIR1-, AFB2-, and AFB3-mCitrine lines and moved the AFB4-mCitrine afb45 picloram response to this figure. AFB1-mCitrine complementation is shown in Figure 6 and AFB5-mCitrine complementation was shown in Prigge et al., 2016.

Reviewer #2:[…]In the subsection “Early-Embryo Defects of the tir1afb235 and tir1afb12345 mutant lines” when the authors detail the observed segregation and types of defects observed during embryo development, they are rather unprecise. There are no exact number or percentage of observed defects in n embryos of what genotype/s. However, this is part is an essential part of the manuscript (and in the title). Could the authors provide quantitative data for: "roughly one quarter of the embryos from each line lacked cotyledons…", for "a rate close to the expected 1/16 ratio" and "expected number of progeny"?

This information is now provided in the text.

If I am correct (from data provided in Figure 1—figure supplement 2), TIR1 and AFB2 genes sit on the same chromosome. Therefore, the phenotype segregation on tir1/+ afb2/+ afb345 might be deviant from the classical Mendelian segregation for 2 alleles, as TIR1 and AFB2 loci may be linked. This is relevant because the embryo phenotypes are dependent on these 2 alleles. This concerns data on Figure 3 and Figure 3—figure supplement 2 (transmission of the sextuple mutant pollen). In the latter one, could you provide the number of crosses analysed per genotype? Some n are rather low (13, 19, 22), which may affect the allelic segregation per genotype. It should be clarified that you are looking at the segregation of the sextuple homozygous offspring versus the rest. If I am not mistaken those sextuple embryos are lethal and this can then affect the segregation analysis. Can you please indicate what tissue was genotyped in this analysis?

TIR1 and AFB2 are 44 cM apart (AGI RI map), and the reviewer is correct that the expected recombination rate (~35% using Kosambi mapping function) would skew the expected proportions. The interpretation is further complicated by the fact that we cannot be certain whether the alleles were still linked in cis rather than in trans in the F4 generation plant used for both crosses. (They were in cis in the F1).

Because of this complication and to increase the statistical power, we replaced this whole experiment with one using reciprocal crosses between Col-0 WT and the sextuple mutant hemizygously complemented with the TIR1/AFB5/AFB2 transgene. The transmission of the transgene could be detected by resistance to basta herbicide in the F1, so larger numbers of F1s could be easily scored. In addition, the 50% expected transmission rather than 25% further enhanced the statistical power. The new Figure 3—figure supplement 2 shows that the sextuple mutant without the transgene was transmitted to nearly 50% of the F1s in both direction of crosses (p = 0.81 and 0.43).

Regarding the tissue used for genotyping in the previous experiment, the F1s were all viable due to the WT alleles for each gene from the Col-0 parent.

One last point concerns the discussion on the expression pattern changes of marker genes in the mutant embryos. I was wondering if there is any information about the expression dependency of those markers on auxin. Is the expression of those genes, auxin-responsive? This is, for example, the case for PIN1, a target of MP (Schlereth et al., 2010; Robert et al., 2015). This would in part explain their absence of expression in the tir1afb235 mutant embryos. PIN1 is absent in the inner cells in the mutant embryos, very similar to what was observed in taa1 tar1 tar2 and yuc1 yuc4 yuc 10 yuc 11 embryos (Robert et al., 2013) when auxin is not provided to the MP-dependent auxin signalling pathway. As for PIN7 switching to the embryo versus the suspensor, this might be the consequence of PIN1 absence and genetic redundancy: PIN7 would take over PIN1 expression domain as it was observed for PIN4 in pin7 embryos (Vieten et al., 2005). At least, this discussion is also missing a conclusive remark.

Both PIN1 and NTT are regulated by MP, and this is now mentioned in the Discussion.

Reviewer #3:In this paper, the authors have analysed a large number of mutant combination in order to reveal the contribution of TIR/AFB auxin receptors to plant development. They have focused on phenotypes of the rosette and the inflorescence, seed number, root growth, and early embryos. The major outcome is that TIR/AFB auxin receptors have largely overlapping functions. Although I feel sympathetic for the efforts the authors have undertaken (keeping track of so many mutant combination itself is worthwhile to note), unfortunately little novel mechanistic insight has been obtained. The authors provide interesting speculations in the Discussion part, such as whether AFB1 might sequester auxin, but these ideas are not experimentally addressed. Altogether, I think this paper provides an important reference point for future work in the auxin field, but at this stage I am not sure whether it contains sufficient novel insight for a broader audience.

We disagree with this reviewer. The original manuscript provides important new insight into the function of the TIR1/AFB proteins, in particular we demonstrate that the TIR1/AFB proteins exhibit a surprising level of functional overlap despite the long time since their divergence. The striking exception is AFB1, which has experienced subfunctionalization so that it is now specialized for the rapid auxin response in the root. We also clearly demonstrate, for the first time, a role for the rapid response in gravitropism.